# Non-coding variability at the *APOE* locus contributes to the Alzheimer's risk

Xiaopu Zhou[1], Yu Chen[1,2,3], Kin Y. Mok[1,4], Timothy C.Y. Kwok[5], Vincent C.T. Mok[6], Qihao Guo[7], Fanny C. Ip[1,2], Yuewen Chen[1,2,3], Nandita Mullapudi [1], Alzheimer's Disease Neuroimaging Initiative, Paola Giusti-Rodríguez [8], Patrick F. Sullivan[8,9,10], John Hardy[4], Amy K.Y. Fu[1,2], Yun Li[8,11] & Nancy Y. Ip[1,2]

Alzheimer's disease (AD) is a leading cause of mortality in the elderly. While the coding change of *APOE*-ε4 is a key risk factor for late-onset AD and has been believed to be the only risk factor in the *APOE* locus, it does not fully explain the risk effect conferred by the locus. Here, we report the identification of AD causal variants in *PVRL2* and *APOC1* regions in proximity to *APOE* and define common risk haplotypes independent of *APOE*-ε4 coding change. These risk haplotypes are associated with changes of AD-related endophenotypes including cognitive performance, and altered expression of *APOE* and its nearby genes in the human brain and blood. High-throughput genome-wide chromosome conformation capture analysis further supports the roles of these risk haplotypes in modulating chromatin states and gene expression in the brain. Our findings provide compelling evidence for additional risk factors in the *APOE* locus that contribute to AD pathogenesis.

[1] Division of Life Science, State Key Laboratory of Molecular Neuroscience and Molecular Neuroscience Center, The Hong Kong University of Science and Technology, Clear Water Bay, Kowloon, Hong Kong, China. [2] Guangdong Provincial Key Laboratory of Brain Science, Disease and Drug Development, HKUST Shenzhen Research Institute, 518057 Shenzhen, Guangdong, China. [3] The Brain Cognition and Brain Disease Institute, Shenzhen Institutes of Advanced Technology, Chinese Academy of Sciences, Shenzhen-Hong Kong Institute of Brain Science-Shenzhen Fundamental Research Institutions, 518055 Shenzhen, Guangdong, China. [4] Department of Molecular Neuroscience, University College London Institute of Neurology, London WC1N 3BG, UK. [5] Therese Pei Fong Chow Research Centre for Prevention of Dementia, Division of Geriatrics, Department of Medicine and Therapeutics, The Chinese University of Hong Kong, Shatin, Hong Kong, China. [6] Gerald Choa Neuroscience Centre, Lui Che Woo Institute of Innovative Medicine, Therese Pei Fong Chow Research Centre for Prevention of Dementia, Division of Neurology, Department of Medicine and Therapeutics, The Chinese University of Hong Kong, Shatin, Hong Kong, China. [7] Department of Neurology, Huashan Hospital, Fudan University, 200040 Shanghai, China. [8] Department of Genetics, University of North Carolina, Chapel Hill, NC, USA 27599. [9] Department of Medical Epidemiology and Biostatistics, Karolinska Institute, SE-171-77 Stockholm, Sweden. [10] Department of Psychiatry, University of North Carolina, Chapel Hill, NC, USA 27599. [11] Department of Biostatistics and Department of Computer Science, University of North Carolina, Chapel Hill, NC, USA 27599. A full list of consortium members appears at the end of the paper. Correspondence and requests for materials should be addressed to N.Y.I. (email: boip@ust.hk)

A lzheimer's disease (AD), a progressive age-related neurodegenerative disorder, is the most common type of dementia and a leading cause of mortality in the elderly. Its prevalence is increasing rapidly with the aging population worldwide[1]. However, its underlying pathological mechanism remains unclear. Over the last few decades, various genetic risk factors for late-onset AD (LOAD) have been identified, including common non-coding variants with low penetrance (odds ratios = 1.05–1.30)[2]. In particular, the *APOE* locus tagged by coding variant *APOE-ε4*, is unequivocally the most significant genetic risk factor for AD[3,4]. While other AD risk variants have also been identified in this region, including *TOMM40* poly-T variation[5–8], *APOE-ε4* is believed to be the only genetic factor that accounts for the risk effect exerted by the *APOE* locus[9].

Apolipoprotein E (ApoE), the lipoprotein encoded by *APOE*, serves as a major lipid carrier in the brain[10]. *APOE* has three isoforms—*APOE-ε2*, *APOE-ε3*, and *APOE-ε4*—defined by combinations of two coding risk mutations (rs429358 and rs7412). *APOE-ε3* is predominant in the general population, while *APOE-ε2* is less common and exerts a protective effect against LOAD. On the other hand, *APOE-ε4* has been identified as a strong AD genetic risk factor, with odds ratios of 1.78–9.93 across different studies or ethnic groups[11–13], and has been reported to modulate brain amyloid-beta (Aβ) burden, tau protein level[14,15], neuronal activity[16,17], immune status[18,19], blood–brain barrier integrity[20] and longevity[21,22]. Thus, *APOE* plays critical roles in both aging and human diseases.

Emerging studies suggest that *APOE-ε4* does not fully explain the AD risk conferred by *APOE* and the surrounding regions[23–26]. Indeed, recent genome-wide association studies (GWAS) for AD conducted in Chinese[27] and European populations[28] have identified leading risk variants in this region, specifically located in the *APOC1* or *PVRL2* loci. Moreover, while individual risk variants residing in non-coding regions exhibit small effect sizes for disease risk, a combination of risk alleles from multiple variants results in aggregate effects, thus contributing to a higher disease risk. Hints of the presence of AD risk haplotype structures in the *APOE* locus have been identified[29,30], although our understanding of these haplotypes has been restricted by traditional genotyping methods (i.e., genotyping array or Sanger sequencing). Thus, there might be additional AD risk variants or haplotype structures in the *APOE* locus that can modulate the risk effects and function of *APOE-ε4* or exert their effects independently. Hence, it is vital to comprehensively analyze AD-associated genetic structures, as well as risk variants in this region in order to better understand the pathological basis of AD and aid the translation of such findings into clinical practice, namely patient stratification and therapeutic development in a genotype-specific manner.

Here, to dissect the complex AD-associated genomic signature within the extended *APOE* region and its contribution to the disease, we perform fine-mapping analysis based on whole-genome sequencing (WGS) and imputed array data from Chinese and non-Asian AD cohorts. We demonstrate the existence of AD risk haplotypes in the *PVRL2* and *APOC1* regions that exert risk effects on AD in an *APOE-ε4* and *APOE-ε2* genotype-independent manner. These risk haplotypes are associated with changes in gene expression, particularly *PVRL2* and *APOE* transcript levels in the brain or blood, and the resultant endophenotypes. Hence, our results collectively suggest that in parallel with the *APOE-ε4* coding risk factor, there are additional genetic risk factors in the *APOE* surrounding regions that can modulate both gene expression and AD-associated phenotypic outcomes, pointing towards new directions for studying the disease mechanisms of AD.

## Results

**AD causal variants in the *PVRL2* and *APOC1* regions.** We recently reported a WGS study of AD in the mainland Chinese population ($n = 1172$; Supplementary Table 1), in which multiple variants located in *APOE* and the surrounding regions exhibited the strongest association with AD[27]. To further investigate the existence of additional risk signals in this region, we conducted fine-mapping analysis in the extended *APOE* region (chr19:45,300,000–45,500,000) using the GATK HaplotypeCaller, which enables the simultaneous detection of SNPs and INDELs in the WGS data of this cohort and an AD cohort from Hong Kong. We applied post-filtering, including controlling for imputation quality (allele dosage DR[2]), allele frequency, and Hardy–Weinberg equilibrium, yielding 682 variants (554 SNPs and 128 INDELs) for subsequent investigation (see Methods section).

To examine whether there are *APOE-ε4*–independent AD risk effects in the *APOE* surrounding regions, we first conducted association analysis among *APOE-ε3* homozygous individuals from the mainland Chinese WGS cohort ($n = 237$ and 288 for the AD and NC groups, respectively) among the 682 obtained variants. A cluster of risk variants near the *APOC1* region was identified. The top signal was observed from rs157592 (effect size $= 1.672$, $p = 3.20 \times 10^{-3}$; Fig. 1a), which indicates that there might be other risk signals in the *APOE* surrounding region in addition to the well-studied *APOE-ε4* risk factor. We subsequently performed an association study for all participants from the mainland AD cohort. Again, the results highlighted the contribution of non-coding variants near *APOC1* to AD pathogenesis (represented by the top candidate rs56131196, effect size $= 0.869$, $p = 1.10 \times 10^{-10}$; Fig. 1b, Table 1). Therefore, we further investigated potential causal variants in this region by performing credible variant analysis through CAVIAR[31]. We identified nine variants with a posterior probability > 10% from three loci—*PVRL2*, *APOE*, and *APOC1* (Table 1)—marked by the following three causal variants with the highest probability: rs11668861 in the *PVRL2* region, rs429358 in the *APOE* region, and rs56131196 in the *APOC1* region (posterior probabilities = 42.5%, 13.9%, and 21.5%, respectively; Fig. 1c, Table 1). These findings suggest the existence of multi-variant effects in *APOE* and the surrounding region, and that the *PVLR2* and *APOC1* loci might contribute to AD pathogenesis in an *APOE-ε4*–independent manner.

Furthermore, we queried the summary statistics from transethnic GWAS summary data reported by Jun et al.[32] from the National Institute on Aging Genetics of Alzheimer's Disease Data Storage Site (NIAGADS). Accordingly, multiple AD-associated variants from the *PVRL2* and *APOC1* loci with $p$-values $< 5 \times 10^{-8}$ were identified in *APOE-ε4* carriers ($n = 12,738$ and 13,850 for AD and NC carrying *APOE-ε4*, respectively; Supplementary Table 2) and in all individuals after adjusting for *APOE-ε4* genotype ($n = 21,392$ and 38,164 for AD and NC, respectively; Supplementary Table 3). Notably, three of the potential causal variants identified in the mainland Chinese WGS dataset (i.e., rs12721051, rs56131196, and rs4420638) remained significant in conditional analyses after adjusting for *APOE-ε4* in the transethnic GWAS results (Supplementary Table 3). Thus, our results indicate the existence of *APOE-ε4*–independent genetic AD risk factors in the *APOE* surrounding region.

**AD risk haplotypes in the *PVRL2* and *APOC1* loci.** To further dissect the AD-associated genetic structure in *APOE* and the surrounding region, we included additional variants (i.e., SNPs and INDELs) that were in LD ($r^2 \geq 0.50$) with the nine causal variants in mainland Chinese WGS dataset, which yielded 33 variants that might reflect the AD-associated genetic signatures in

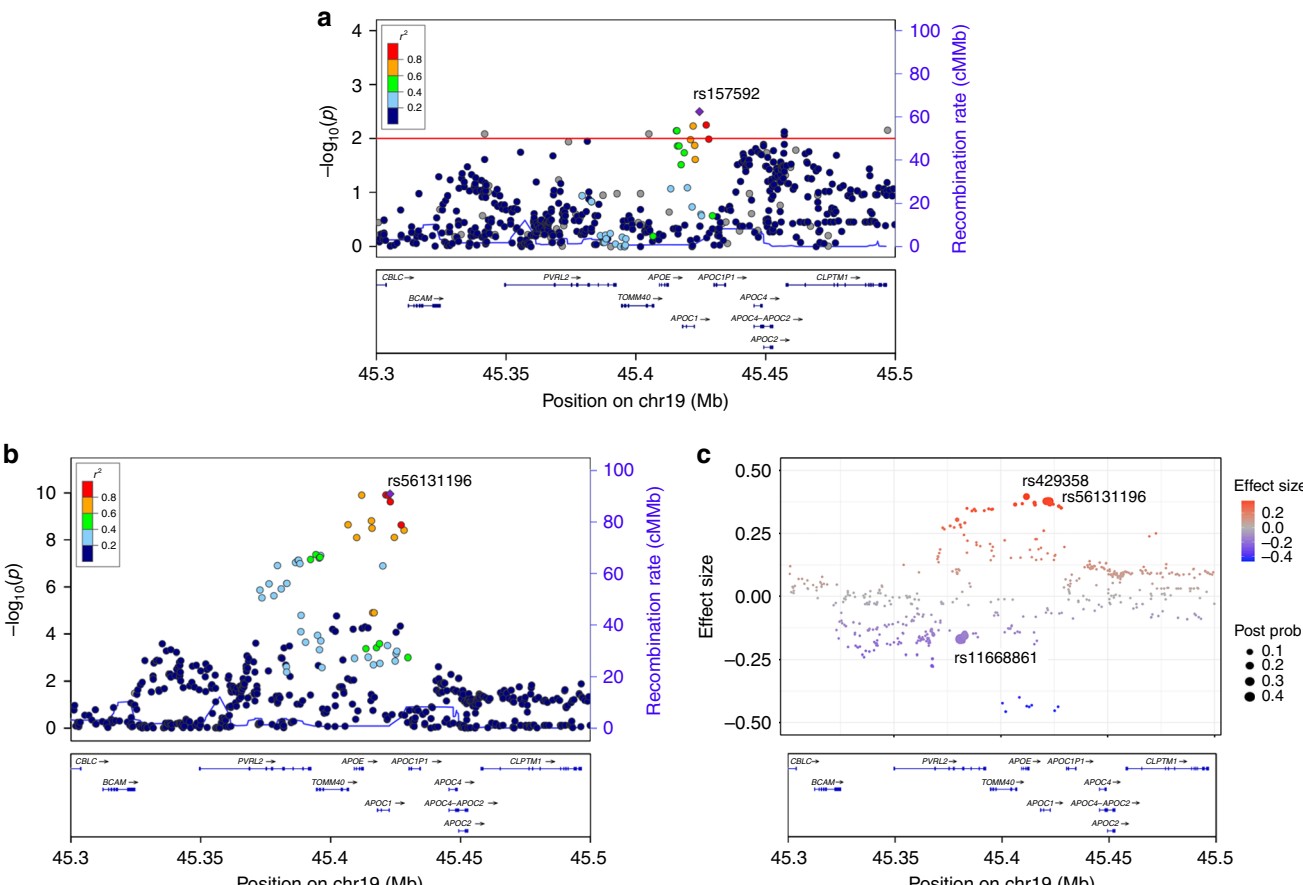

**Fig. 1** Multivariant effects of the *APOE* locus in the Chinese AD cohort. **a** Regional association plot of the AD risk variants in *APOE-ε*3 homozygous subjects. The horizontal red line denotes the *p*-value threshold of 0.01. **b** Regional association plot of the AD risk variants (SNPs and INDELs with frequency ≥ 5%) located in the *APOE* locus. The purple diamond specifies the sentinel variant (with the SNP ID marked in the plot). Dot colors illustrate the LD (measured as $R^2$) between the sentinel variant and its neighboring variants. **c** CAVIAR analysis results for mapping of possible causal variants in the *APOE* locus. Dots represent the variants tested in the *APOE* locus; the *y*-axis and dot color denote the effect size. Dot size corresponds to the posterior probabilities of the variants being the causal variants obtained from CAVIAR analysis, with the sentinel variants located in three loci marked with SNP IDs. AD Alzheimer's disease, CAVIAR causal variants identification in associated regions, cM/Mb centimorgans per megabase, INDELs insertions and deletions, LD linkage disequilibrium, SNP single nucleotide polymorphism, Post Prob posterior probabilities of being the causal variants

**Table 1 Potential causal variants in *APOE* and the surrounding region identified by CAVIAR analysis**

| SNP | BP | Gene | EA | *Beta* | SE | *Z*-value | *p*-value | TF binding | EAF in NC (Mainland/HK/ADNI/ADC/LOAD) |
|---|---|---|---|---|---|---|---|---|---|
| rs11668861 | 19:45380970 | PVRL2 | T | −0.39 | 0.12 | −3.28 | 1.0E−03 | Yes | 0.78/0.79/0.55/0.53/0.54 |
| rs6859 | 19:45382034 | PVRL2 | G | −0.40 | 0.11 | −3.54 | 3.9E−04 | Yes | 0.69/0.71/0.43/0.42/0.42 |
| rs3852860 | 19:45382966 | PVRL2 | T | −0.36 | 0.12 | −3.03 | 2.4E−03 | Yes | 0.76/0.77/0.59/0.58/0.59 |
| rs3852861 | 19:45383061 | PVRL2 | T | −0.34 | 0.12 | −2.87 | 4.1E−03 | Yes | 0.76/0.77/0.59/0.58/0.59 |
| rs429358 | 19:45411941 | APOE | C | 0.91 | 0.14 | 6.44 | 1.2E−10 | No | 0.11/0.11/0.14/0.14/0.21 |
| rs12721046 | 19:45421254 | APOC1 | A | 0.87 | 0.14 | 6.44 | 1.2E−10 | No | 0.13/0.09/0.13/0.13/0.17 |
| rs12721051 | 19:45422160 | APOC1 | G | 0.87 | 0.14 | 6.43 | 1.3E−10 | Yes | 0.13/0.09/0.17/0.17/0.22 |
| rs56131196 | 19:45422846 | APOC1 | A | 0.87 | 0.14 | 6.45 | 1.1E−10 | No | 0.13/0.09/0.19/0.17/0.22 |
| rs4420638 | 19:45422946 | APOC1 | G | 0.85 | 0.13 | 6.34 | 2.4E−10 | No | 0.13/0.09/0.19/0.17/0.22 |

*Note*: CAVIAR analysis results for the major causal variants, defined as a posterior probability ≥ 10%, with a summary for variants frequency in normal control participants from each studied cohort. Regions with transcription factor-binding events annotated by the ENCODE database are marked as "yes" in the "TF binding" column. The last column displayed the effective allele frequencies of corresponding variants in the normal control populations of given cohorts accordingly

*BP* base position in GRCh37 annotation, *Gene* nearest genes, *EA* effect allele, *Beta* effect size, *SE* standard error, *TF* transcription factor, *EAF* effect allele frequency, *NC* normal controls

this region (Supplementary Table 4). Haplotype analysis revealed two major haplotype blocks defined by variants extending from the *PVRL2* and *APOC1* causal variants (Fig. 2a). The stratified LD plots showed that AD patients manifested a distinct genomic structure relative to NC groups, as represented by stronger LD (i.e., larger pairwise $r^2$ values between variants) among risk

variants in the *PVRL2*, *APOE*, and *APOC1* loci, suggesting that these AD risk variants are more likely to coexist in AD (Fig. 2a). We replicated this analysis in the ADNI WGS dataset (*n* = 808) and observed similar LD patterns in AD (Supplementary Fig. 1). Moreover, we identified multiple haplotypes (frequency > 5% in the NC groups) in the *PVRL2* and *APOC1* haplotype blocks in the

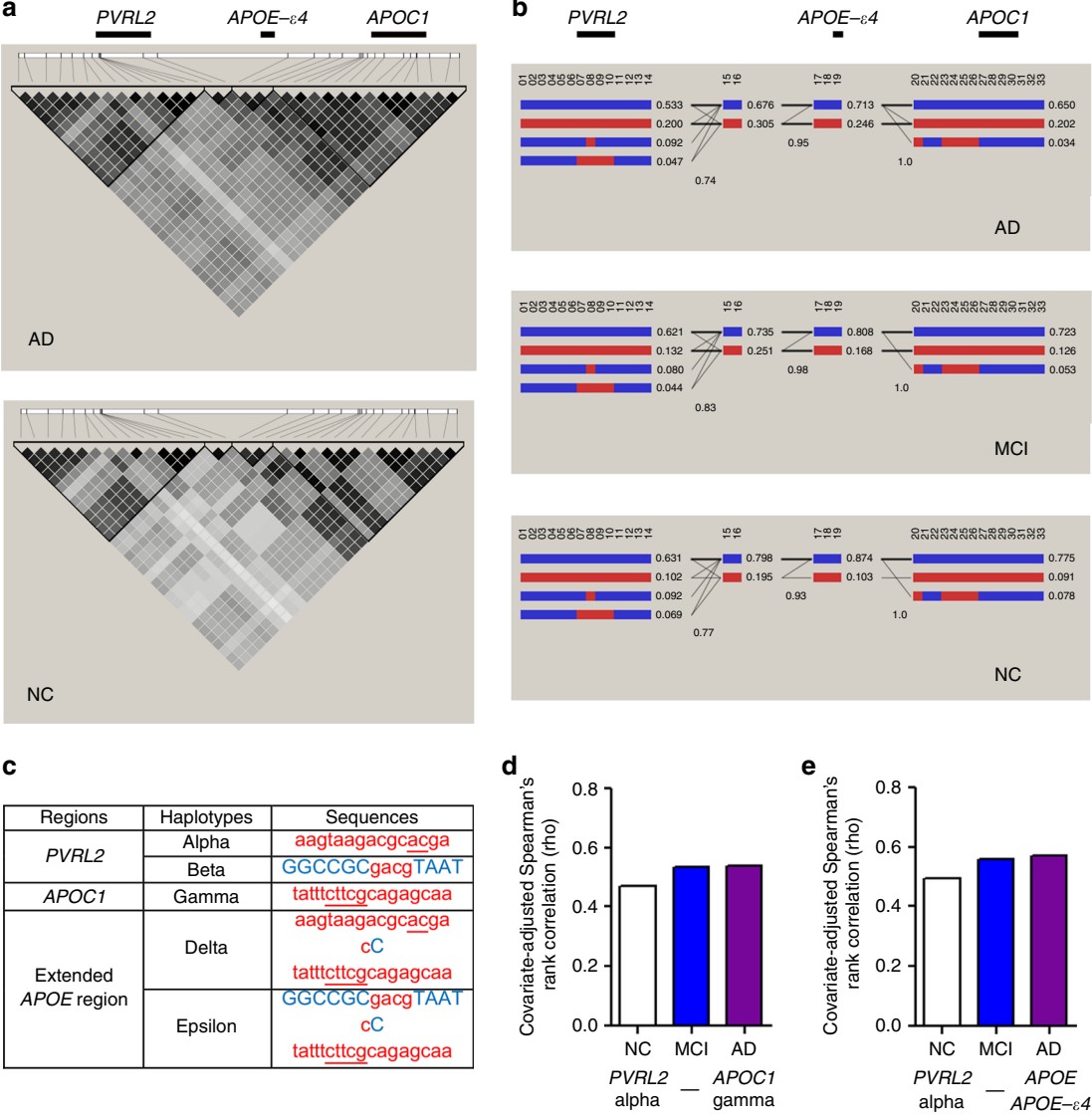

**Fig. 2** Haplotype structure of AD-associated risk variants in the Chinese AD cohort. **a** Pairwise LD plot of the 33 selected variants in LD with the potential risk variants in different phenotypic groups. The color map corresponds to the pairwise $r^2$ values between variants, with nine potential risk variants located in the *PVRL2*, *APOE*, and *APOC1* loci marked at the top panel, respectively. **b** Haplotype analysis of the 33 selected variants among different phenotypic groups. Each column (marked with numbers) represents one of the 33 variants, with red and blue indicating the minor (i.e., AD risk) and major alleles, respectively. Each row represents a particular haplotype defined by a specific combination of major and minor alleles in the given haplotype blocks, with decimals on the right side specifying the frequencies of corresponding haplotypes in the given phenotypic groups. Intersecting lines represent the frequency of associations between two connected haplotypes (thin and thick lines denote associations with frequency > 1% and > 10% in the corresponding groups, respectively). **c** Table summarizing the identified minor haplotypes in *PVRL2*, *APOC1*, and extended *APOE* regions. Letters in uppercase blue or lowercase red denote the major and minor (risk) alleles, respectively; underlined letters highlight INDELs. **d, e** Pairwise correlations between the minor haplotypes of *PVRL2* alpha and *APOC1* gamma or *APOE*-ε4 measured by Spearman's partial rank-order correlation, adjusted for age, gender, and principal components in corresponding phenotypic groups (presented as Spearman's $\rho$ in the y-axis). AD Alzheimer's disease, INDELs insertions and deletions, LD linkage disequilibrium, MCI mild cognitive impairment, NC normal controls

mainland Chinese WGS data (Fig. 2b), particularly the minor haplotypes defined by the minor alleles of all variants within blocks that cover *PVRL2* or *APOC1* gene bodies (i.e., *PVRL2* haplotype alpha and *APOC1* haplotype gamma, respectively; Fig. 2b, c). In addition, these minor haplotypes were enriched and more frequently associated with each other in the MCI and AD groups than the NC group (Fig. 2b); thus, these minor haplotypes might contribute to AD, and there might be extended haplotypes spanning the *PVRL2–APOE–APOC1* region formed by the combination of the abovementioned minor haplotypes from these three genomic regions.

We subsequently performed haplotype inference in a variant pool containing the *PVLR2* and *APOC1* haplotype blocks (comprising 14 variants for each haplotype block), as well as two coding variants representing *APOE* haplotypes (rs429358 and rs7412) by resolving their phased states (as recorded in phased VCF files) at the individual level. Using a partial correlation test controlling for confounding factors, we confirmed that there were more frequent associations between *PVRL2* haplotype alpha and *APOC1* haplotype gamma or *APOE*-ε4 in the AD and MCI groups when compared to the control groups (Fig. 2c–e; Supplementary Table 5); we validated these findings in the ADNI WGS and Hong Kong Chinese WGS cohorts

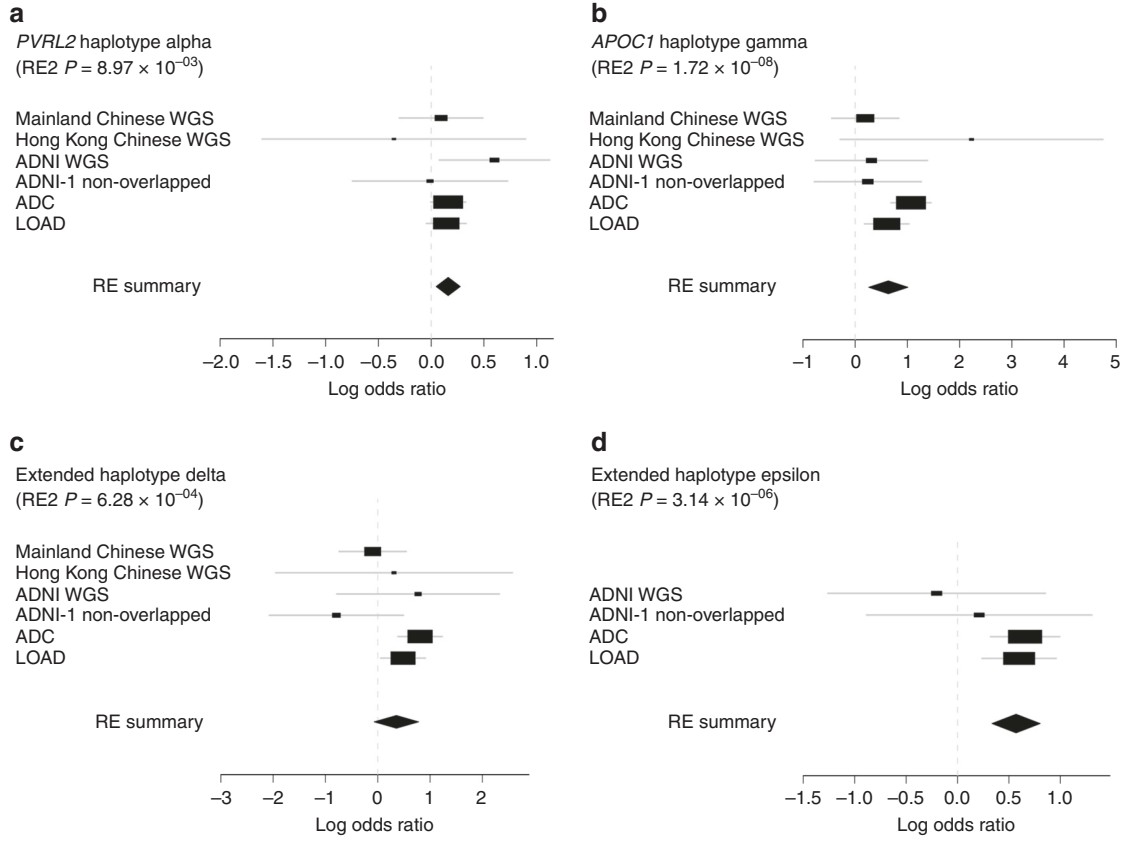

**Fig. 3** Forest plot of haplotypes contributing to AD after controlling for *APOE* genotypes. Forest plot with values of effect size obtained from independent datasets or meta-results denoted by rectangles and diamonds, respectively. For each row representing the independent dataset, lines indicate 95% confidence intervals, and sizes of rectangles are proportional to the weights in the meta-analysis. **a, b** *PVRL2* alpha and *APOC1* gamma haplotypes were associated with AD in an *APOE* genotype-independent manner (*p*-values shown are for Han and Eskin's random effects model). **c, d** Association results of extended minor haplotypes delta and epsilon after controlling for *APOE-ε4* genotypes (*p*-values are for Han and Eskin's random effects model). AD Alzheimer's disease, RE random effects, RE2 Han and Eskin's random effects model

(Supplementary Tables 1, 6, 7). In addition, we confirmed the existence of the minor haplotypes in *PVRL2* and *APOC1* loci (*PVRL2* haplotype alpha, *PVRL2* haplotype beta, and *APOC1* haplotype gamma), as well as *APOE-ε4*–harboring extended haplotypes (haplotypes delta and epsilon; Fig. 2c) defined by the combination of *PVRL2*, *APOE*, and *APOC1* minor haplotypes in non-Asian populations (predominantly Caucasian populations using three array-based AD genetic datasets, ADC, LOAD, and ADNI; Supplementary Tables 1, 8). In summary, we identified *PVRL2* and *APOC1* and *APOE* extended haplotypes, which are potentially associated with AD, located in *APOE* and the surrounding region in the general population.

**APOE-ε4–independent effects of the AD risk haplotypes**. We subsequently used a multivariate model to evaluate the risk effects of the aforementioned minor haplotypes and determine their associations with AD (Supplementary Table 9−11). Meta-analysis highlighted the haplotypes' risk effects for AD, with all meta–*p*-values passing the genome-wide significance threshold ($p < 5 \times 10^{-8}$; Supplementary Table 12). Notably, after controlling for *APOE* genotypes (both *APOE-ε4* and *APOE-ε2*), *PVRL2* haplotype alpha, *APOC1* haplotype gamma, and the two *APOE-ε4*–harboring extended haplotypes (delta and epsilon) still manifested as conferring a significantly elevated risk for AD (Supplementary Tables 11, 12). Meta-analysis summarizing the statistics from all datasets ($n = 7092$ and 4856 for the AD and NC groups, respectively) corroborated the haplotypes' risk effects (meta-*p* < 0.01; Fig. 3a–d, Table 2, Supplementary

Tables 13, 14). Thus, we identified AD-associated haplotypes that encompass *APOC1* and *PVRL2*, and contribute to AD in an *APOE-ε4* genotype-independent manner.

Furthermore, we replicated the above analysis in individuals harboring homozygous *APOE-ε3* alleles. While *APOC1* haplotype gamma was significantly associated with AD (effect size = 2.203, $p = 6.84 \times 10^{-3}$), *PVRL2* haplotype alpha was significantly associated with AD in females in the mainland cohort (effect size = 0.980, $p = 0.038$ in females). The concordant risk effects for *PVRL2* haplotype alpha were observed in females in the ADC (effect size = 0.165, $p = 0.250$) and LOAD (effect size = 0.072, $p = 0.720$) cohorts. Thus, these results further support the risk effect of *PVRL2* haplotypes in AD, especially in females.

**Cross-platform validation of the AD risk haplotypes**. To examine the accuracy of our haplotype-phasing method, we adopted two independent datasets: the mainland Chinese WGS dataset and the ADNI WGS datasets, both of which have the WGS and array data available. Both datasets indicated that our analysis method can achieve more than 95% accuracy (Supplementary Fig. 2, Supplementary Tables 15, 16) for haplotypes with a frequency > 5%. Furthermore, we obtained sequencing data from the Ashkenazim son–father–mother trio from the Personal Genome Project[33], which comprises high-coverage (~300×) Illumina short-read data and long-read PacBio data (~50× coverage), and confirmed the existence of *PVRL2* haplotype alpha and *APOC1* haplotype gamma in the general population (HG003, the father, carries both haplotypes; Supplementary Table 17). We

**Table 2 Meta-analysis of AD-associated haplotypes after controlling for *APOE* genotypes**

| Haplotypes | Study # | Beta (RE) | SD (RE) | *p*-value (RE2) | $I^2$ | Q | *p*-value (Q) | Tau$^2$ |
|---|---|---|---|---|---|---|---|---|
| Haplotypes in the *PVRL2* region | | | | | | | | |
| aagtaagacgcacga | 6 | 0.161 | 0.059 | 8.97E−03 | 0.00 | 3.68 | 5.96E−01 | 0.00 |
| GGCCGCgacgTAAT | 6 | 0.059 | 0.070 | 6.97E−02 | 40.12 | 8.35 | 1.38E−01 | 0.01 |
| GGCCGCTGcgTAAT | 5 | −0.098 | 0.096 | 3.66E−01 | 0.00 | 3.68 | 4.51E−01 | 0.00 |
| Haplotypes in the *APOC1* region | | | | | | | | |
| tatttcttcgcagagcaa | 6 | 0.635 | 0.193 | 1.72E−08 | 42.43 | 8.69 | 1.22E−01 | 0.08 |
| tGGttcttcgcGCGAATG | 6 | −0.345 | 0.316 | 3.40E−01 | 0.00 | 2.60 | 7.62E−01 | 0.00 |
| Extended haplotypes | | | | | | | | |
| aagtaagacgcacga cC tatttcttcgcagagcaa (ε4) | 6 | 0.356 | 0.218 | 6.28E−04 | 45.62 | 9.20 | 1.02E−01 | 0.11 |
| GGCCGCTGTTTAAT cC tatttcttcgcagagcaa (ε4) | 6 | 0.312 | 0.259 | 4.10E−04 | 51.06 | 10.22 | 6.93E−02 | 0.17 |
| GGCCGCgacgTAAT cC tatttcttcgcagagcaa (ε4) | 4 | 0.570 | 0.120 | 3.14E−06 | 0.00 | 2.73 | 4.35E−01 | 0.00 |

*Note*: Summary metrics from association results, controlling for *APOE*-ε4 and *APOE*-ε2 genotypes obtained from different AD cohort data, were subjected to METASOFT for meta-analysis. A random effects (RE) model based on inverse-variance-weighted effect size was applied to estimate summary-level effect size (*Beta*) and standard deviation. Han and Eskin's random effects (RE2) model was applied to estimate the significance level, accounting for possible heterogeneity across populations
*AD* Alzheimer's disease, *Beta* effect size, *SE* standard deviation, *RE* random effects model, *RE2* Han and Eskin's random effects model, *$I^2$* I-squared heterogeneity statistic, *Q* Cochrane's Q-statistic, *Tau$^2$* Tau-squared heterogeneity estimator of Der Simonian–Laird

further performed target-region PacBio sequencing for nine lymphoblastoid cell lines harboring target haplotypes (zero, one, or two copies of extended haplotype delta). All nine cell lines exhibited concordant phasing results, despite a minor inconsistency in the detection of small INDELs (Supplementary Tables 18, 19). Thus, we demonstrated the existence of AD risk haplotype structures in the general population, as well as the accuracy of our detection method for both the WGS and imputed array data.

**Effects of AD risk haplotypes on endophenotypes**. We subsequently examined the effects of the identified risk haplotypes on cognitive performance, brain volumetric imaging, and levels of cerebrospinal fluid (CSF) and plasma biomarkers from ADNI dataset by using a multivariate model integrating information for the *PVRL2*, *APOE*, and *APOC1* risk haplotypes. *PVRL2* haplotype alpha was associated with worsening cognitive performance as assessed by the Everyday Cognitive Scale ($p = 2.27 \times 10^{-4}$; total score reported by study partners; Fig. 4a, Supplementary Table 20), individual memory performance ($p = 2.54 \times 10^{-2}$ and $2.27 \times 10^{-4}$ for observations assessed by participants and study partners, respectively; Fig. 4b, Supplementary Tables 21, 22), individual linguistic performance ($p = 4.91 \times 10^{-2}$; Supplementary Table 23), and planning ($p = 6.23 \times 10^{-2}$; Supplementary Table 24). Accordingly, *PVRL2* haplotype alpha was associated with decreased brain volume including whole brain volume ($p = 3.33 \times 10^{-2}$; Supplementary Table 25), middle temporal lobe volume ($p = 3.29 \times 10^{-2}$; Supplementary Table 26), and particularly the volume of the hippocampus, which plays key roles in memory-associated behaviors ($p = 2.14 \times 10^{-2}$; Fig. 4c, Supplementary Table 27). The haplotype was also associated with changes in total Aβ$_{1-42}$ plasma level (FDR = 0.009; Fig. 4d, Supplementary Table 28) and a reduction in intercellular adhesion molecule 1 (ICAM-1) level in CSF (FDR = 0.054; Fig. 4e, Supplementary Table 29). In contrast, *APOC1* haplotype gamma was associated with the plasma levels of free Aβ$_{1-40}$ (FDR < 0.001; Fig. 4f, Supplementary Table 28) and monocyte-chemotactic protein 3 (MCP3, also called chemokine ligand 7 [CCL7]; FDR = 0.040; Fig. 4g, Supplementary Table 30) in a dose-dependent manner. These results indicate that the identified *PVRL2* and *APOC1* risk haplotypes affect a variety of clinical and biochemical indexes including cognitive performance (especially memory function), brain volume, and plasma and CSF biomarkers—all in an *APOE*-ε4–independent manner (Supplementary Fig. 3). This

corroborates our previous findings and indicates that these risk haplotypes may play critical roles in the AD pathogenesis.

**Association of risk haplotypes with gene expression changes**. Given that non-coding variants are potentially associated with the regulation of gene expression, we examined whether the variants in the identified risk haplotypes are located within regulatory regions in the human genome. The UCSC Genome Browser[34] suggested that some of these variants are located in transcription factor-binding regions (Supplementary Fig. 4). Thus, the identified *PVRL2* and *APOC1* risk haplotypes tagged by those variants might exert biological effects by modulating the expression of nearby genes. Corroborating the potential association between genetic variants and gene regulatory functions, genotype–expression association analysis using GTEx data for individual variants in *APOE* and the surrounding regions ($n = 96$; Supplementary Figs. 5–8, Supplementary Tables 31–34) showed that *PVRL2* variants exerted a significant local regulatory effect on blood *PVRL2* transcript level (rs60389450, $p = 8.82 \times 10^{-34}$; Supplementary Fig. 5, Supplementary Table 31), whereas *PVRL2* and *APOC1* variants exhibited a distal modulatory effect on *APOE* transcript levels in brain tissue (meta-$p = 1.30 \times 10^{-5}$ and $1.08 \times 10^{-5}$ for *PVRL2* variant rs519113 and *APOC1* variant rs60049679, respectively; Supplementary Fig. 7, Supplementary Table 33). Given that *PVRL2* variant rs519113 resides in the variant pool defining the *PVRL2* haplotypes (Supplementary Table 4), the identified *PVRL2* AD-risk haplotypes might influence *APOE* expression level in the brain.

We subsequently performed genotype–expression association analysis with the GTEx dataset, which revealed that *PVLR2* minor haplotypes were associated with reduced blood *PVRL2* transcript level ($p = 1.77 \times 10^{-2}$ and $6.95 \times 10^{-6}$ for *PVRL2* haplotypes alpha and beta, respectively; Fig. 5a, Supplementary Table 35). We observed the same associations in *APOE*-ε3 homozygous carriers ($p = 2.41 \times 10^{-2}$ and $1.05 \times 10^{-4}$ for *PVRL2* haplotypes alpha and beta, respectively; Supplementary Table 36). In the brain, *PVRL2* haplotype alpha and *APOC1* haplotype gamma exhibited concordant associations with increased *APOE* and *APOC1* transcript levels (alpha: effect size = 0.347 and 0.273, meta-$p < 0.05$; gamma: effect size = 0.559 and 0.518, meta-$p < 0.001$; for *APOE* and *APOC1* brain transcript levels, respectively; Fig. 5b, Supplementary Table 41), suggesting that the identified risk haplotypes have a distal regulatory effect on *APOE* expression in the brain.

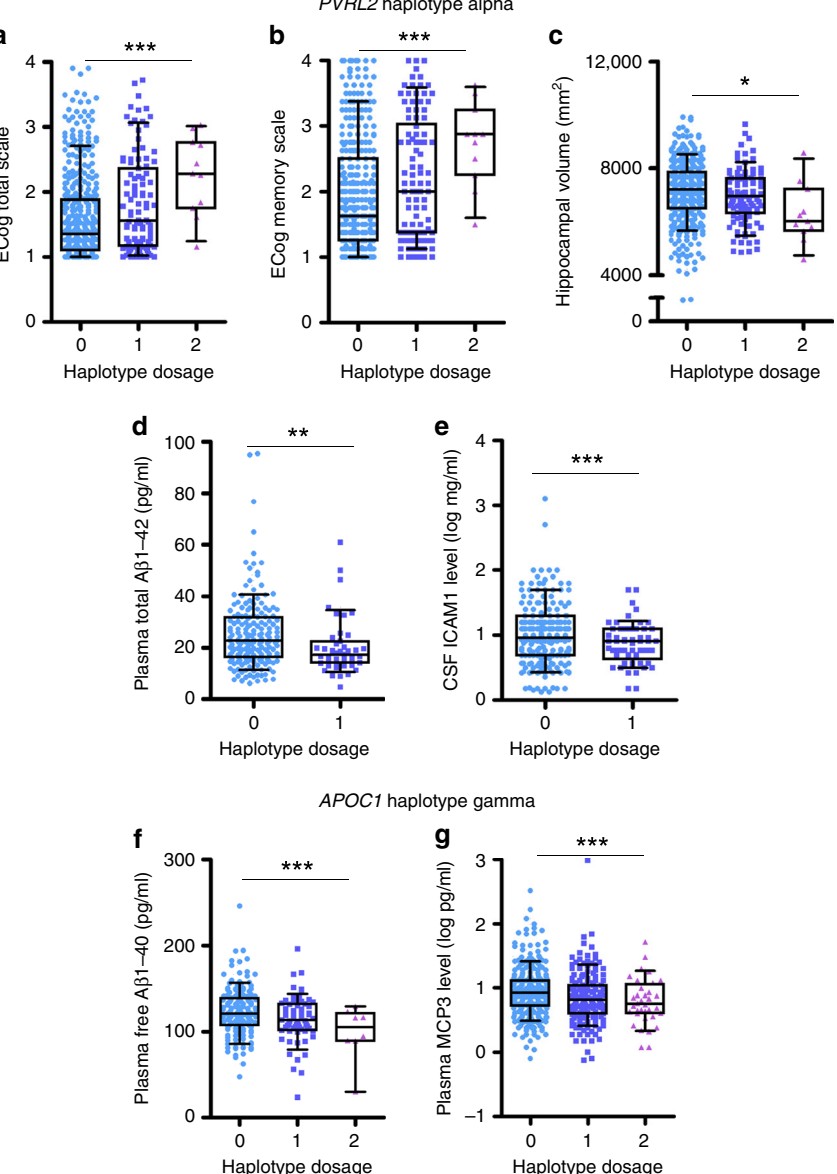

**Fig. 4** Functional implications of *PVRL2* and *APOC1* haplotypes in an *APOE*-ε4–independent manner. **a–e**. Associations between *PVRL2* minor haplotype alpha, and cognitive performance and biomarker expression in an *APOE*-ε4-independent manner. **a, b** Associations between *PVRL2* alpha haplotype dosage with (**a**) cognitive performance indicated by total ECog score (scored between 0−4; higher scores represent more severe disability in functioning) reported by study partners ($n = 527$, $T = 3.71$, ***$p < 0.001$, $Beta = 0.25$) and (**b**) memory performance indicated by ECog memory score reported by study partners ($n = 527$, $T = 3.60$, ***$p < 0.001$, $Beta = 0.29$). **c** Association between *PVRL2* alpha haplotype with hippocampal volume ($n = 1{,}121$, $T = -2.31$, *$p < 0.05$, $Beta = -165.60$ [mm$^2$]). **d, e** Associations between *PVRL2* alpha haplotype with (**d**) total A$\beta_{1-42}$ in plasma ($n = 226$, $T = -3.098$, **$p < 0.01$, $Beta = -4.113$ [pg/mL]) and (**e**) ICAM-1 in cerebrospinal fluid; $n = 298$, $T = -3.361$, ***$p < 0.001$, $Beta = -0.199$ [log ng/mL]). Individuals harboring two copies of haplotypes were not included due to the small samples size. **f, g** Association between *APOC1* gamma haplotype with levels of (**f**) plasma free A$_{\beta1-40}$ ($n = 226$, $T = -4.823$, ***$p < 0.001$, $Beta = -40.231$ [pg/mL]) and (**g**) plasma MCP3 (CCL7) ($n = 537$, $T = -3.665$, ***$p < 0.001$, $Beta = -0.229$ [log ng/mL]). A$\beta$ amyloid-beta, ECog everyday cognition. Data are presented in box plots, with boxes extending from the 25th to 75th percentiles and whiskers specifying the 10th and 90th percentiles; the line in the middle of the box denotes the median

Interestingly, *APOE*-ε4 was associated with a consistent decrease of *TOMM40*, *APOE*, and *APOC1* transcript levels in the brain (effect size = −0.370, −0.392, and −0.444, respectively, meta-$p < 0.01$; Fig. 5b, Supplementary Table 41), implying that *APOE*-ε4 has a suppressive effect on the nearby genes. Moreover, we observed a concordant increase in blood and brain transcript levels of *APOE* with increasing age (effect size = 0.014 and 0.011, $p < 0.001$; Fig. 5a, b, Supplementary Tables 35, 41). These results further suggest that aging affects gene expression, particularly *APOE* transcript levels in the brain and blood.

To further understand the regulatory mechanisms underlying the effects of different risk alleles or haplotypes, we examined changes in the levels of genes transcript(s) that carry specific alleles (allelic imbalance) or specific isoforms. *PVRL2* is mainly expressed as three isoforms in blood: ENST00000252485.4, ENST00000591585.1, and ENST00000252483.5. The first two harbor a UTR that covers variant rs6859, which is a causal variant from the *PVRL2* risk haplotypes. Association analysis between *PVRL2* haplotypes and blood *PVRL2* isoform levels revealed that haplotype beta exerted its suppressive effect on the blood *PVRL2*

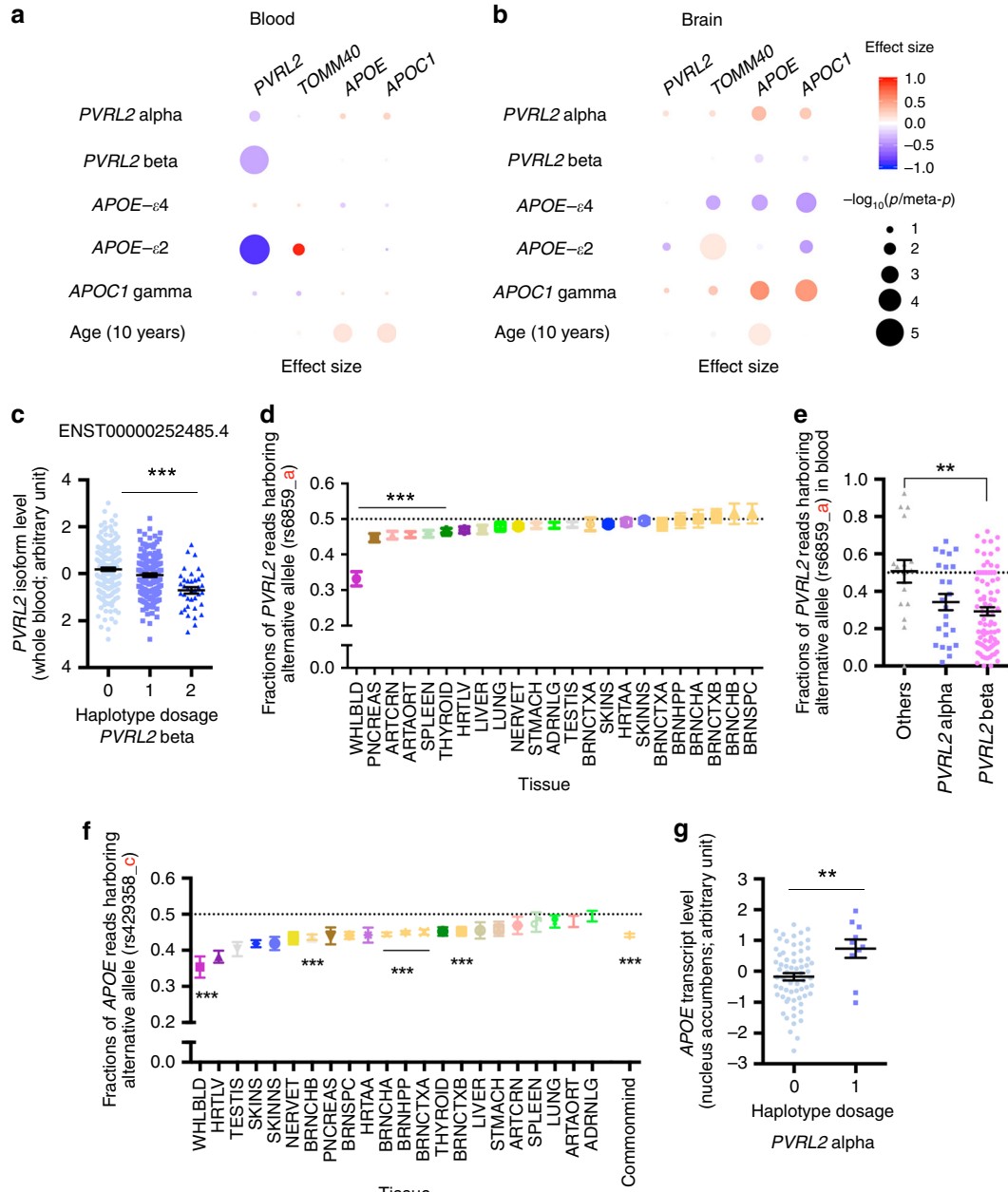

**Fig. 5** Modulatory effects of AD-associated haplotypes in *APOE* and the surrounding region on the expression of nearby genes. **a, b** Dot plots showing the haplotype–expression association results of the AD-associated haplotypes in *PVRL2*, *APOE*, and *APOC1* and their nearby genes in (**a**) blood and (**b**) the brain. Dot color and size represent effect size and significance level (*p* or meta-*p* values), respectively. **c** Association between *PVRL2* beta haplotype with transcript level of blood *PVRL2* isoform (ENST00000252485.4) ($n = 365$, $T = -5.470$, ***$p < 0.001$, $Beta = -0.449$). **d** Allelic imbalance of *PVRL2* variant rs6859 across multiple tissues. One-sample *t*-test (***$p < 0.001$). Data were obtained from the GTEx dataset. **e** Association between *PVRL2* minor haplotypes and transcripts of *PVRL2* with variant rs6859 in blood ($n = 124$, $T = -3.218$, **$p < 0.01$, $Beta = -0.209$, for beta haplotype against the major haplotype). **f** Allelic imbalance of *APOE* variant rs429358 across multiple tissues. One-sample *t*-test (***$p < 0.001$ for representative results). Data were obtained from the GTEx and CommonMind datasets. **g** *PVRL2* haplotype alpha was associated with changes of brain *APOE* transcript level in individuals not carrying an *APOE*-ε4 allele (nucleus accumbens, $n = 67$ or 10 for non-*APOE*-ε4 carrying individuals harboring 0 or 1 copies of *PVRL2* haplotype alpha, respectively; $T = 2.963$, **$p < 0.01$, $Beta = 0.943$, for alpha haplotype against the major haplotype). Data are plotted as mean ± SEM in **c**, **e**, and **g**. Tissue abbreviations: ADRNLG adrenal gland, ARTAORT artery-aorta, ARTCRN artery-coronary, BRNCHA brain-cerebellum, BRNCHB brain-cerebellar hemisphere, BRNCTXA brain-cortex, BRNCTXB brain-frontal cortex (BA9), BRNHPP brain-hippocampus, BRNSPC brain-spinal cord (cervical c-1), HRTAA heart-atrial appendage, HRTLV heart-left ventricle, LIVER liver, LUNG lung, NERVET nerve-tibial, PNCREAS pancreas, SKINNS skin-not sun exposed (Suprapubic), SKINS skin-sun exposed (Lower leg); SPLEEN spleen, STMACH stomach, TESTIS testis, THYROID thyroid, UTERUS uterus, WHLBLD whole blood

isoforms with a UTR that covers variant rs6859 (effect size = $-0.449$ and $-0.426$, $p < 1 \times 10^{-4}$ for ENST00000252485.4 and ENST00000591585.1 vs. ENST00000252483.5: effect size = $-0.184$, $p = 3.03 \times 10^{-2}$; Fig. 5c, Supplementary Fig. 9, Supplementary Table 37). Moreover, analysis of the GTEx dataset revealed an allelic imbalance (i.e., a decrease of risk allele-harboring transcript) of rs6859 in multiple tissues (Fig. 5d), with the strongest effect in blood ($n = 124$, average fraction of minor alleles = 0.332; $p < 0.0001$; Fig. 5d, Supplementary Table 38). Droplet digital PCR (ddPCR) verified the allelic imbalance of rs6859 in blood *PVRL2* transcript (average fraction of minor alleles in blood RNA = 0.302; Supplementary Fig. 10). By querying the cis-eQTL data obtained from the eQTLGen Consortium[35], rs6859 was again significantly associated with blood *PVRL2* transcript level ($n = 29,726$, $p = 5.38 \times 10^{-300}$, Z-score = $-37.02$). Furthermore, *PVRL2* haplotype beta was associated with a reduced fraction of rs6859 minor allele-harboring transcript in blood ($n = 124$, effect size = $-0.209$, $p = 0.002$; Fig. 5e, Supplementary Table 39), indicating that haplotype beta may have modulatory effects on the transcriptional activity of *PVRL2* in blood in an allele-specific manner.

Other than causing an amino acid mutation in ApoE protein, *APOE* variant rs429358 exhibited allelic imbalance in multiple tissues, demonstrating a suppressive effect of variant rs429358 on the expression of the risk allele-harboring transcript (Fig. 5f). Unlike *PVRL2* rs6859, we also observed an allelic imbalance of *APOE* rs429358 in brain tissues (average fraction of minor alleles = 0.442, $p < 0.0001$ in CommonMind; Fig. 5d, f, Supplementary Table 38), which corroborates the aforementioned suppressive effect of *APOE*-ε4 on brain *APOE* transcript level (Fig. 5b, f). In contrast, *PVRL2* haplotype alpha and *APOC1* haplotype gamma were associated with an elevated *APOE* transcript level in the brain (meta-$p < 0.01$; Supplementary Table 40), especially in individuals without *APOE*-ε4 (*PVRL2* haplotype alpha: effect size = 0.271, meta-$p = 2.68 \times 10^{-2}$; *APOC1* haplotype gamma: effect size = 1.284, meta-$p = 1.43 \times 10^{-8}$; Fig. 5g, Supplementary Table 41) and in individuals with homozygous *APOE*-ε3 alleles (*PVRL2* haplotype alpha: effect size = 0.247, meta-$p = 3.02 \times 10^{-2}$; Supplementary Table 42). These results suggest that dysregulated *APOE* expression is involved in AD pathogenesis in parallel with the dysfunctions conferred by *APOE*-ε4 allele.

**Physical interactions of haplotype regions in the brain.** To examine the possible mechanisms that contribute to the regulatory effects of the *PVRL2*, *APOE*, and *APOC1* risk haplotypes on the expression of nearby genes in brain tissues, we adopted Hi-C data from two datasets: one comprising pooled samples from both adult and fetal human brains[36], and the other comprising Hi-C data from the germinal zone (GZ) and cortical plate (CP) of the fetal brain[37]. We identified multiple interaction hotspots in *APOE* and the surrounding regions including regions that cover the risk haplotypes, i.e., the *APOE* risk haplotype region (45,410–45,420 kb), *PVRL2* risk haplotype region (45,370–45,380 kb), and *APOC1* risk haplotype region (45,430–45,440 kb). We also identified multiple interaction hotspots in other non-haplotype regions including the *PVRL2* promoter region (45,330–45,340 kb), *PVRL2* region (45,360–45,370 kb; ~2.8 kb upstream of the *PVRL2* risk haplotype), *PVRL2–TOMM40* region (45,390–45,400 kb; ~6.8 kb downstream of the *PVRL2* risk haplotype), and *APOC1P1* region (45,440–45,450 kb).

Regarding the interaction hotspots that cover the risk haplotypes, the *APOE* risk haplotype region exhibited physical interactions with the *PVRL2–TOMM40* and *APOC1P1* regions (FDR < 0.05; Fig. 6, Supplementary Tables 43, 44). Meanwhile, regarding the *PVRL2* and *APOC1* risk haplotypes associated with gene expression changes in the brain (Fig. 5b), the *APOC1* risk

haplotype region interacted with the *PVRL2–TOMM40* region (FDR < $1 \times 10^{-9}$ for the adult and fetal brains; Fig. 6, Supplementary Table 43), and the *PVRL2* risk haplotype region interacted with the *PVRL2* promoter region in the adult brain (FDR < 0.001; Fig. 6, Supplementary Tables 43, 44). Interestingly, distal interactions with the risk haplotype region ($p < 0.05$) covering a broad genomic region were observed in both fetal and adult brain tissues (Supplementary Figs. 11, 12), implying that non-coding haplotypes might have broad modulatory effects on nearby genes. These observations suggest the complexity of chromatin states that might contribute to the regulation of transcriptional activity, prompted the urgency for the further investigation of associated chromatin structure changes in the brain during the aging or dementing stage.

**Functional implications of the AD risk haplotype variants.** In line with the genotype–expression association analysis and observed chromatin interaction events, the identified non-coding risk variants likely function through modulating local transcript factor or microRNA binding. We first queried the non-coding risk variants to determine their potential functions. Several non-coding risk variants, including rs6859 and rs483082, as well as one INDEL, rs11568822, were co-localized with histone modifications and/or transcription factor-binding regions (Supplementary Fig. 13). Subsequent electrophoretic mobility shift assay for genomic regions harboring those variants confirmed their binding capability with nuclear protein (Supplementary Fig. 14), implying that these non-coding variants play roles in the modulation of transcription factor binding.

Furthermore, MicroSNiPer[38] database query of rs6859, which is located in the UTR of *PVRL2* transcript, returned microRNA candidates including miR-595, miR-636, and miR-1825—all of which might bind to the rs6859 region (Supplementary Table 45). These binding events were further assessed by independent in silico alignment using miRanda (Supplementary Table 46). Specifically, miR-595 was predicted to only interact with the major G allele of rs6859 and not the minor A allele (Supplementary Tables 46, 47). This suggests that rs6859 might also affect the *PVRL2* transcript level through the modulation of microRNA binding events at its UTR in parallel with transcription factor binding at the DNA level.

**Haplotype prevalence is heterogeneous among ethnic groups.** To corroborate the observed differences in haplotype frequency across the Chinese and non-Asian datasets (Supplementary Table 8), we assessed individual haplotype frequency using the 1000 Genomes Project phase 3 data ($n = 2,504$) and stratified the individuals into five "super-populations." The results show heterogeneity among ethnic groups (Fig. 7, Supplementary Table 48). Regarding *APOE*, *APOE*-ε4 was most frequent in the African population (frequency = 0.267) and was less frequent in the East Asian population than the European population (frequency = 0.086 vs. 0.155, respectively), whereas *APOE*-ε2 was more frequent in the East Asian population than the European population (0.100 vs. 0.063, respectively). The prevalence of *PVRL2* haplotype alpha was similar between the East Asian and European populations (0.102 and 0.103, respectively). However, *PVRL2* haplotype beta and *APOC1* haplotype gamma were much less frequent in the East Asian population than the European population (haplotype beta = 0.081 vs. 0.318, haplotype gamma = 0.066 vs. 0.111, respectively). As for long-range AD risk haplotypes, haplotype delta was most frequent in the East Asian population (0.043 vs. 0.016, 0.027, 0.021, and 0.002 in the South Asian, American, European, and African populations, respectively), whereas haplotype epsilon was most frequent in the European population (0.059 vs. < 0.001, 0.008, 0.006,

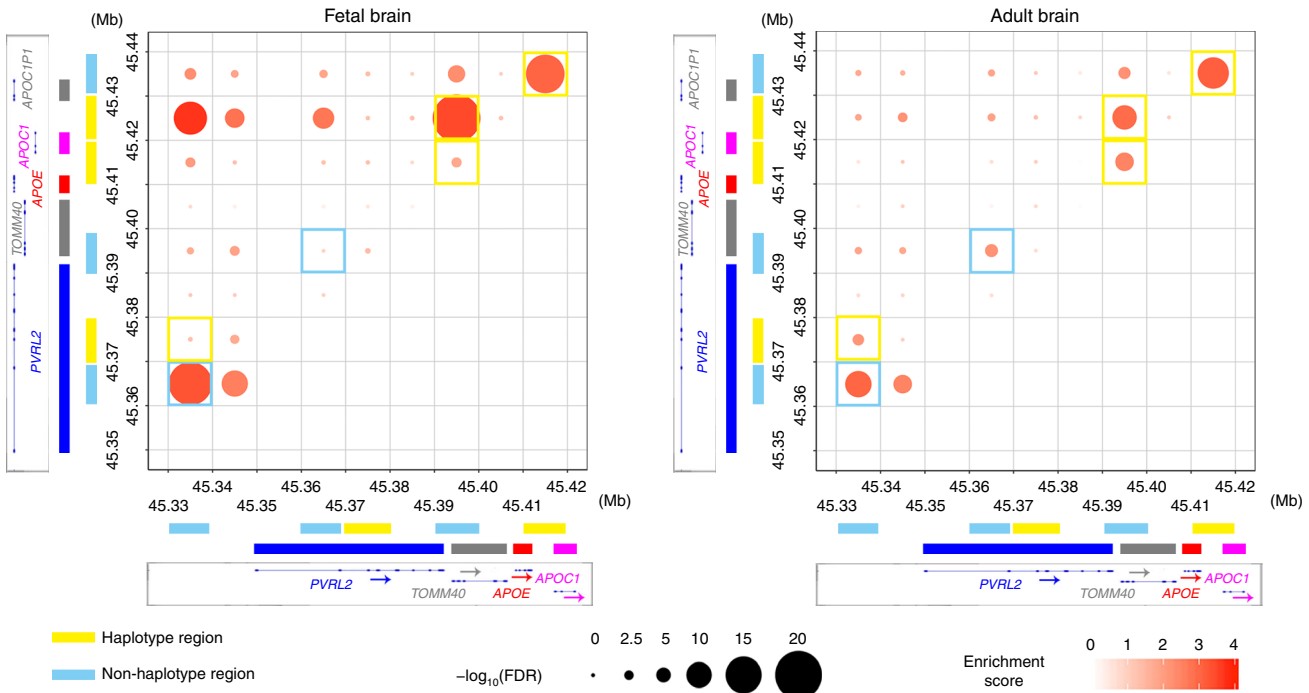

**Fig. 6** Chromatin interaction analysis showing the physical interactions between the *PVRL2*, *APOE*, and *APOC1* regions in fetal and adult human brain tissues. Chromatin interaction events were measured by Hi-C assay. Dot plots show the physical interaction events in and near the *APOE* region (19:45,330–45,440 kb) with a bin size of 10 kb. X and y-axes denote the genomic coordinates, with the corresponding gene body marked on the side (blue, red, and pink bars denote the gene body regions of *PVRL2*, *APOE*, and *APOC1*, respectively). Genomic regions that cover the risk haplotypes are denoted in yellow, and haplotype regions are denoted in cyan. The color intensity of dots represents the interaction strength of the corresponding pair of genomic regions (the enrichment score was calculated by dividing the observed number of contact events by the expected number of contact events; dark colors indicate strong interactions). Dot size corresponds to statistical significance ($-\log_{10}$ of the FDR); larger dots indicate higher confidence of the observed interaction. Hi-C results obtained from fetal (left panel) and adult (right panel) brain are shown. Interaction hotspots located in the regions that cover risk haplotypes and non-haplotype regions are bordered by yellow and cyan, respectively. For both groups, three replicates were pooled for the analysis. Mb megabases in GRCh37 coordinates, FDR false discovery rate

and 0.002 in the East Asian, South Asian, American, and African populations, respectively). These findings suggest the existence of possible divergent mechanisms of AD pathogenesis among ethnic groups and demonstrate how ethnic diversity might influence the relative risk of a disease at the population level.

## Discussion

Here, we report a comprehensive analysis of *APOE* and the surrounding region using WGS data, which revealed specific AD-associated genetic structures. Our haplotype analysis identified *PVRL2* and *APOC1* minor haplotypes that exhibit independent risk effects for AD in parallel with *APOE*-ε4, as well as long-range AD risk haplotypes defined by the combination of *PVLR2*, *APOE*, and *APOC1* risk haplotypes that exhibit stronger risk effects than *APOE*-ε4 alone. We also demonstrated that the AD risk haplotypes are associated with endophenotypes. The regulatory effects of the risk haplotypes on the brain transcript levels of *APOE* and its nearby genes, together with the identification of chromatin interaction hotspots in and near the *APOE* risk loci, all support involvement of the identified genetic risk factors in the *APOE* locus play pathological roles in AD in parallel with *APOE*-ε4.

Most previous genetic studies identified genetic risk factors at the single-variant level[2,27,28]. However, individual genetic variants can only explain a small proportion of the variations of complex traits (e.g., phenotypic consequences of diseases or gene expression), which are largely due to polygenetic effects (i.e., combined effects of multiple common variants)[39,40]. Corroborating this notion, we have identified AD risk haplotypes in *APOE* and the surrounding region that harbor functional variants (Table 1). In

particular, the identified minor haplotypes in the *PVRL2* and *APOC1* regions exhibit *APOE*-ε4–independent AD risk effects. Thus, our fine-mapping work extends the current understanding of the *APOE* locus as a risk factor for AD beyond the well-studied *APOE*-ε4 to a more complex genomic structure and its associated regulatory mechanisms. In particular, we showed that the risk haplotypes potentially exert biological impacts through modulating endophenotypes including memory performance, hippocampal volume, proteomic biomarkers in CSF and plasma, and transcriptome signatures in the brain and blood. Thus, these results demonstrate the functional implications of the risk effects of the non-coding variants/haplotypes from the macroscale to the microscale. Their roles in gene expression are further supported by the chromatin interaction events of the *APOE* locus in human brain tissues, as well as the risk variant-dependent regulation of microRNA and nuclear protein binding (Supplementary Fig. 14, Supplementary Tables 46, 47). These results are vital for more comprehensive analyses of phenotype-associated genomic structures in AD risk loci or the contribution of polygenic effects to AD-associated phenotypes. These findings might also facilitate AD mechanistic studies or the development of risk prediction or intervention strategies in a genotype-aware manner.

Regarding the identified risk loci, *PVRL2* and *APOC1*, in the *APOE* surrounding region, *PVRL2* encodes poliovirus receptor-related 2, which is a glycoprotein and a component of the plasma membrane that serves as an entry point for herpesvirus and pseudorabies virus[41]. While it was recently reported that levels of herpesvirus, HHV-6A, and HHV-7 are elevated in postmortem AD brains compared to normal brains[42], whether the regulation

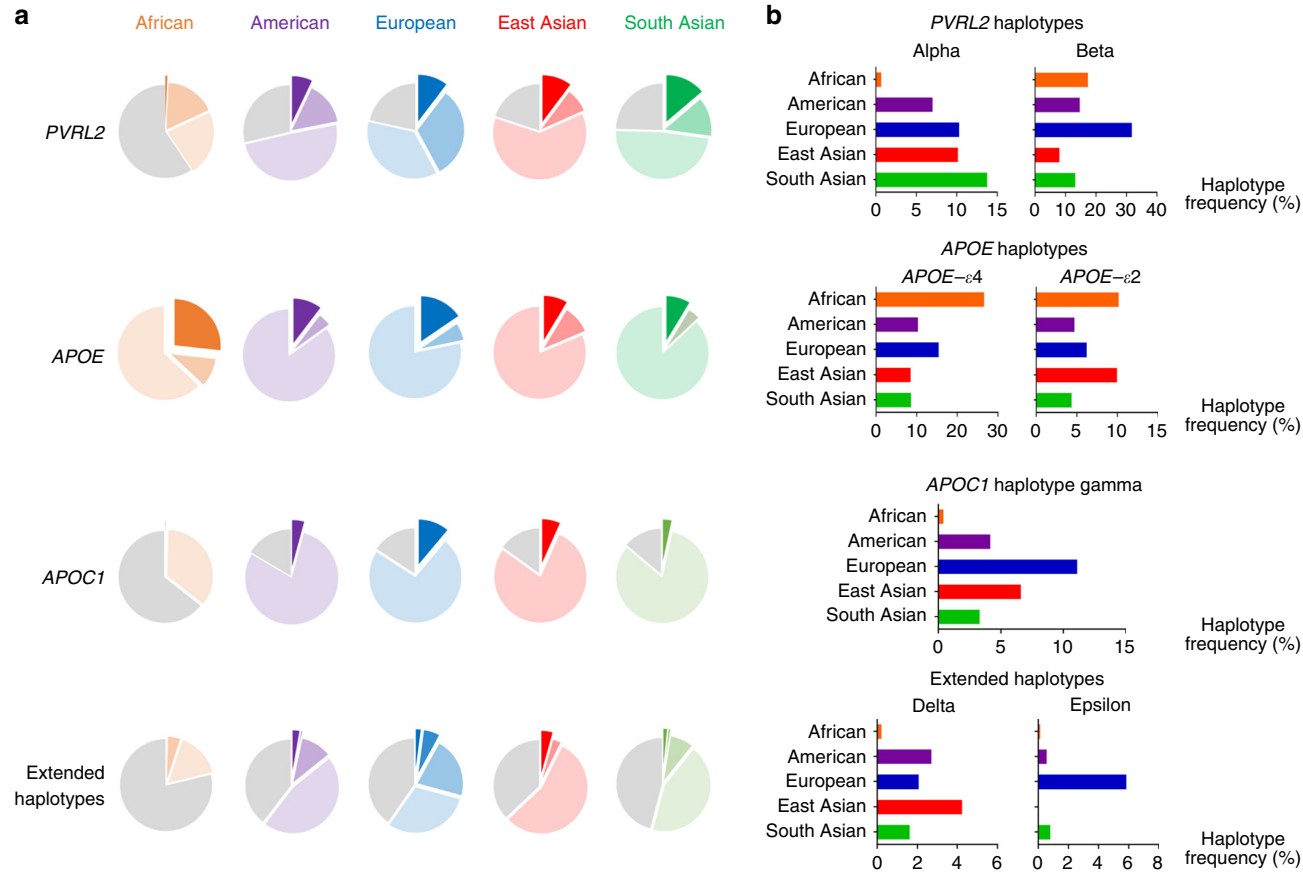

**Fig. 7** Heterogeneity of the prevalence of risk haplotypes in *APOE* and the surrounding region among populations. Data were derived from the 1000 Genomes phase 3 whole-genome sequencing dataset ($n = 2504$, comprising 661 African, 347 American, 503 European, 504 East Asian, and 489 South Asian genomes). **a** Major and minor haplotypes among populations. The proportions of minor haplotypes are shown as exploded areas in dark colors; areas with light colors denote the proportions of major haplotypes (i.e., the most frequent ones), and gray areas indicate the proportions of all other haplotypes for the corresponding locus. Exploded areas denote the phenotype-associated minor haplotypes for comparison across ethnic groups, with detailed frequencies shown in the right panel. **b** Frequencies of minor haplotypes across different ethnic groups, with the *x*-axis denoting the haplotype frequencies across super-populations

of *PVRL2* expression, specifically in blood, affects viral entry in AD patients requires further study. Meanwhile, *APOC1* encodes apolipoprotein C1, which is mainly involved in lipoprotein metabolism and might inhibit the ApoE-mediated uptake of very-low-density lipoprotein particles[43]. Thus, it is important to examine whether altered *APOC1* expression regulates ApoE functions such as ApoE-associated Aβ clearance in AD states.

*APOE* or *APOE*-ε4 transcript levels in the brain might also be crucial for the pathogenesis of AD. Alterations of *APOE* signatures have been observed in AD brain tissues[26,44,45]. Meanwhile, non-coding AD genetic risk factors might mediate their effects by modulating gene expression in specific cellular contexts[46,47]. The present study showed that the identified *PVRL2* and *APOC1* risk haplotypes are potentially associated with elevated brain *APOE* transcript level, which is consistent with the changes in brain *APOE* level during aging; this suggests that a higher brain *APOE* (or *APOE*-ε4) level is associated with the risk of disease pathogenesis. Notably, AD transgenic mouse model(s) exhibit higher hippocampal *APOE* transcript levels than corresponding wild-type mice (Supplementary Fig. 15a). Moreover, *APOE* transcript levels are strongly correlated with hippocampal plaque pathology in AD transgenic mice ($R^2 > 0.70$; Supplementary Fig. 15b). In addition, recent studies show that *APOE* expression is elevated in disease-associated microglia in an AD transgenic mouse model[48] and microglia with a neurodegenerative phenotype[49]; these results collectively implicate elevated *APOE* level in inflammatory response, AD disease onset, and AD progression. Thus, in

addition to *APOE*-ε4 genetic risk factors, elevated brain *APOE* level might be critical for the pathogenesis of AD. Furthermore, our analysis provides additional clues regarding the suppressive effects of *APOE*-ε4 on *APOE* expression in the brain after controlling for the genetic content in the *PVRL2* and *APOC1* regions. Thus, further investigation is required to determine how *APOE*-ε4 mediates the regulatory roles of *APOE* expression.

In conclusion, we identified AD risk haplotypes with putative biological effects that confer AD risk. Our findings suggest the existence of alternative disease mechanisms involving non-coding variants in the *APOE* surrounding regions, which act in parallel with the well-studied *APOE*-ε4 risk factor. Our results further demonstrate the complexity of the genetic basis associated with AD pathogenesis, which might result in aggregate risk effects from both intrinsic factors such as mutant proteins defined by coding mutations, the local and distal regulation of gene expression by genomic contents, as well as extrinsic factors including aging, viral infection, and ethnic variation. Further investigations aiming to further dissect the underlying mechanism of AD will be of great importance for the development of effective diagnostics and therapeutics.

## Methods

**Mainland Chinese AD WGS cohort.** The AD cohort comprised 1172 participants recruited from Huashan Hospital, Fudan University, Shanghai, including 477 AD patients (AD group), 253 with mild cognitive impairment (MCI group), and 442 corresponding age-matched and gender-matched cognitive normal controls

(NC group)[27]. AD patients were diagnosed according to the recommendations of the National Institute on Aging and the Alzheimer's Association workgroup[50,51] and had an onset age ≥ 50 years. Patients with MCI were diagnosed according to the Peterson criteria[52]. We excluded individuals with any significant neurologic disease or psychiatric disorder. This study was approved by the Ethics Committee of Huashan Hospital, the Hong Kong University of Science and Technology (HKUST) and the HKUST Shenzhen Research Institute. All subjects provided written informed consent for both study enrollment and sample collection.

**Hong Kong Chinese AD WGS cohort.** A total of 208 participants, including 109 with AD and 99 age-matched NCs, were recruited from the Specialist Outpatient Department of the Prince of Wales Hospital, the Chinese University of Hong Kong. AD patients (age: 65–93 years) were diagnosed based on the American Psychiatric Association's Diagnostic and Statistical Manual of Mental Disorders, Fifth Edition (DSM-5)[53]. All AD patients underwent subsequent neuroimaging assessment (i.e., magnetic resonance imaging, MRI), as well as cognitive and functional tests. All participants including AD patients and NCs were examined for cognitive normality using the Mini-Mental State Examination or Montreal Cognitive Assessment test[54,55]. The phenotypes of the participants were determined on the basis of the latest diagnostic records (until April, 2018). This study was approved by the Prince of Wales Hospital, the Chinese University of Hong Kong, and HKUST. All participants provided written informed consent for both study enrollment and sample collection. Blood genomic DNA was extracted and subjected to WGS using Truseq Nano DNA HT Sample Preparation Kit (Illumina). Prior to association testing, two samples (one AD and one NC) were filtered out owing to relatedness (PLINK[56] IBD estimation), leaving 206 samples ($n = 108$ and 98 for AD and NC groups, respectively) for downstream analysis. Please refer to *Supplementary Methods* in the *Supplementary Information* for more detailed descriptions.

**Other study cohort and datasets.** Additional AD cohorts were included in the present analysis, including (i) genotype, transcriptome, brain volumetric and biomarker data from the Alzheimer's Disease Neuroimaging Initiative (ADNI) database (adni.loni.usc.edu/); (ii) genotype and phenotype data from the National Institute on Aging Alzheimer's Disease Centers Cohort (ADC) (phs000372.v2.p1); and (iii) genotype and phenotype data from the Late Onset Alzheimer's Disease (LOAD) Family Study (phs000168.v2.p2). In addition, for transcriptome and allele-specific analysis, genotype and transcriptome data from (iv) Genotype-Tissue Expression (GTEx) project (phs000424.v6.p1) and (v) CommonMind Consortium Data were included and analyzed. Please refer to *Supplementary Methods* in the *Supplementary Information* for detailed descriptions.

**Variant detection in *APOE* and the surrounding region.** To simultaneously obtain single nucleotide polymorphisms (SNPs), as well as insertions and deletions (INDELs) in *APOE* and the surrounding region (chr19:45,300,000–45,550,000) from the WGS data generated separately in two Chinese cohorts (mainland Chinese and Hong Kong Chinese WGS cohorts), the Genome Analysis Tool Kit[57–59] (GATK, v3.4–46-gbc02625) HaplotypeCaller was adopted for variant calling. Variant recalibration was subsequently applied for SNPs and INDELs using VariantRecalibrator (truth sensitivity thresholds of 90% and 99.9% for INDELs and SNPs, respectively). Top variants ranked by VQSLOD score that passed the sensitivity thresholds were retained for genotype refinement and phasing using Beagle[60,61] (r1399). Post-filtering was applied for allele-dosage $R^2$ ($DR^2 > 0.30$), minor allele frequency (MAF > 5%), and Hardy–Weinberg Equilibrium ($p > 1 \times 10^{-5}$) for all SNPs and INDELs, yielding 682 variants (554 SNPs and 128 INDELs). Please refer to *Supplementary Methods* in *Supplementary Information* for detailed information.

**Covariates adjustments in association analysis.** In general, for all statistical analyses, age, gender, and the top five principal components (PCs) were included as covariates separately within individual cohort. Principal components analysis was conducted using the PLINK[56] (version 1.9)–*pca* function with the pruned (–*indep-pairwise 50 5 0.2*) variants with an MAF > 5%. For Chinese AD cohorts, the genome-wide variant calling was obtained using Gotcloud pipeline with genotyping refinement performed by Beagle[60,61] (r1399) (*nthreads* = 24, *phase-its* = 30, *impute-its* = 15; Please refer to *Supplementary Methods* for more detailed information). For ADNI biomarker data, phenotypic labels were included as covariates. For ADNI brain volumetric data, the analysis was further adjusted for the type of MRI platform, analytical software, and individual intracranial volume.

**Association test at the single variant level.** We used PLINK[56,62] (version 1.9) for logistic regression analysis of SNPs and INDELs with an MAF > 5% in *APOE* and the surrounding region (chr19:45,300,000–45,500,000), controlling for age, gender, and the top five ancestry PCs; 682 variants passed these filters and were included in the analysis (–*hwe 1E-05*,–*maf 0.05*). We subjected the PLINK association results (i.e., *Z*-score) with pairwise linkage disequilibrium (LD) information (i.e., the $r^2$ matrix obtained from PLINK–*matrix* with–*r* function) to CAVIAR[31] (Causal Variants Identification in Associated Regions) analysis (version 2.0.0) to estimate the potential causal variants within the *APOE* locus indicated by the posterior probability of being the causal variants.

**Multivariate regression analysis for haplotype function.** Multivariate regression analysis was performed to estimate the effects of specific haplotypes on phenotype or gene expression because of the existence of multiple haplotypes in the study cohort. An $N \times (M + 1)$ matrix was generated for a cohort comprising $N$ individuals (in rows) and $M$ detected haplotypes with frequencies > 1% or > 5% (in columns), with cells storing a value of 0, 1, or 2, representing the harboring of 0, 1, or 2 copies of specific haplotypes, respectively. In the last column ($M + 1$th column), the haplotype counts for haplotypes with a frequency < 1% were summed and annotated as "others" to ensure the sum of each row equaled 2. Major haplotypes (usually $Hap_1$ denoted by all major alleles, which is the most frequent in the population) were excluded in the regression model during the association test. Thus, the effect sizes (*beta*) from the model above were estimated with respect to the major haplotype.

To further control the effects from other haplotype regions, the genetic dosages of minor haplotypes from all haplotype blocks were included in the present models with minor revision of above model. See *Supplementary materials and methods* for a detailed description about the analytical model.

**Association test and meta-analysis of candidate haplotypes.** Minor haplotypes with frequencies > 1% were included in the multivariate logistic regression model using the R *glm* function from the *stats* package. Analyses were performed separately for the *PVRL2*, *APOC1*, and long-range haplotypes defined by the combination of *PVRL2*, *APOE*, and *APOC1* haplotypes. The analyses were controlled for *APOE* genotype by incorporating the genotype dosages of *APOE*-ε4 and *APOE*-ε2 into the model. The effect size and standard errors (SE) obtained from the logistic regression were subjected to METASOFT[63] to generate the meta-analysis results using a random effects (RE) model, with statistical significance estimated by Han and Eskin's random effects model (RE2).

**Association test for haplotypes on endophenotypes.** A multivariate model jointly taking haplotype information from the *PVRL2*, *APOE*, and *APOC1* loci was adopted to assay the haplotype effects on cognitive score, brain volumetric data, and ADNI biomarker levels using robust regression (R *lmrob* from the *robustbase* package) with appropriate covariate adjustments. For ADNI biomarker data, Bonferroni adjustment was applied for the association test of individual biomarkers to correct for the multiple tests on haplotypes, whereas the false discovery rate (FDR) was calculated for individual haplotypes across all biomarkers. Adjustments were performed using the *p.adjust* function from the R *stats* package.

**Association test for variants/haplotypes on gene expression.** GTEx data comprising the transcript levels of *PVRL2*, *TOMM40*, *APOE*, and *APOC1* (rank-based inverse normal transformed by the R *rntransform* function from the *GenABEL*[64] package) together with imputed genotype data for variants with an MAF > 5% located in non-repetitive regions (UCSC RepeatMasker in hg19 coordinate) were included in the genotype–phenotype association test using PLINK, with age, gender, and the top five PCs as covariates. To estimate the variant effects for all tissues or 13 brain tissues, meta-analysis was conducted using the *rma* in the R package *metafor*[65] (*method* = "HE," *test* = "knha"), taking effect sizes and standard errors from the PLINK results. For haplotype data, association tests were conducted using the multivariate model, jointly including *PVRL2*, *APOE*, and *APOC1* haplotype information using the robust regression model. Among the brain tissues, the cerebellum, cerebellar hemisphere, and spinal cord were excluded from the meta-analysis conducted by METASOFT using the RE model, with statistical significance estimated by the RE2 model for haplotype effects in brain tissues. For ASE data in GTEx data, robust regression was applied to test associations. For ASE in the GTEx and CommonMind datasets, one-sample *t*-tests were applied to examine allele imbalance under the null hypothesis of balanced expression (i.e., the fraction of reads carrying minor alleles = 0.5 as the theoretical values) using GraphPad Prism 6 (GraphPad Software Inc.) at an α level of 0.05.

**Chromatin interaction analysis in brain tissues.** Two high-throughput chromosome conformation capture (Hi-C) datasets were adopted to investigate the chromatin organization in *APOE* and surrounding regions. The first dataset comprised anterior temporal cortex samples from three adults of European ancestry with no psychiatric disorders, as well as cerebral cortex samples from three fetal brains at a gestational age of 17–19 weeks[36]. All samples were free from large structure variations (>100 kb), and easy Hi-C (eHi-C) methods were adopted for library construction, sequencing, and data analysis[66]. The second dataset comprised data generated from three paired germinal zone and cortical plate fetal brain samples[37]. Briefly, for both datasets, pooled or individual data were mapped to human reference genome (hg19) using *BWA mem* or Bowtie[67]. The uniquely mapped paired-end reads passing quality controls were further binned into 10-kb bin resolution contact matrices, and the data were then subjected to Fit-Hi-C[68] and FastHiC[69,70] to assess chromatin interaction events in this region. The FDR was further calculated to identify interaction hotspots.

**Data visualization.** The GWAS results were visualized using Locuszoom[71] plots, with LD and *p*-values obtained from the WGS data. The CAVIAR results and heatmap for haplotype effects were visualized using the *ggplot* function in the *ggplot2* R package. LD and haplotype structures were plotted using Haploview. Bar

charts, dot plots, box plots, and line charts were generated using GraphPad Prism 6 (GraphPad Software Inc). Forest plots for meta-analysis were generated using ForestPMPlot[72]. Pie charts were generated using Excel 2017 (Microsoft).

**Web Resources**. For R, see [https://www.r-project.org/]; for ADNI, see [http://adni.loni.usc.edu]; for 1000 Genomes project phase 3 data, see [http://www.internationalgenome.org/data]; for GTEx Portal, see [https://gtexportal.org/home/] (raw data under dbGaP phs000424.v6.p1); for CommonMind, see [https://www.synapse.org/#!Synapse:syn2759792/wiki/69613]; for UCSC genome browser, see [https://genome.ucsc.edu/cgi-bin/hgTracks]; for Mouseac dataset, see [http://www.mouseac.org]; for MicroSNiPer, see [vm24141.virt.gwdg.de/services/microsniper/]; for eQTLGen, see [http://www.eqtlgen.org/cis-eqtls.html].

**Reporting summary**. Further information on research design is available in the Nature Research Reporting Summary linked to this article.

## Data availability

The summary-level statistics for the association results in APOE and the nearby regions, raw PacBio sequencing data generated in lymphoblastoid cell lines, and variant calling results for PacBio sequencing data are available at [http://iplabdatabase.ust.hk/zhou_et_al_2019/APOE_data.html]. The Hi-C data can be found on the PGC website, the HUGIn online database, and Gene Expression Omnibus (GEO) with accession number GSE116825. The National Institute on Aging—Late Onset Alzheimer's Disease Family Study (LOAD) raw data were accessed in dbGaP phs000168.v2.p2; the Alzheimer's Disease Genetics Consortium (ADGC) Genome Wide Association Study—NIA Alzheimer's Disease Centers Cohort (ADC) raw data were accessed in dbGaP at phs000372.v2.p1; the Alzheimer's Disease Neuroimaging Initiative (ADNI) dataset were accessed at ADNI database [http://adni.loni.usc.edu/]. For mainland WGS data, the genetic information at individual level can be shared upon approval after reviewed by Human Genetics Resources Administration of China (HGRAC). For Hong Kong WGS data, raw sequencing data can be found on [http://iplabdatabase.ust.hk/CND/AD_registry_study.html]. The consent that was given from individual participants stated that the research content will be kept private under supervision of the hospital and research team. Thus, the data will be available and shared in the form of a formal collaboration basis; application of data sharing and project collaboration will be processed and reviewed by a Reviewing Panel hosted at HKUST. Researchers may further contact [sklneurosci@ust.hk] for the details for data sharing and project collaboration in this study. The source data underlying Supplementary Figs. 14a and 14b are provided as a Source Data file.

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

## Acknowledgements

We thank Dr. Yu Pong Ng, Dr. Kwok Wang Hung, Ka Chun Lok, Cara Kwong, Yuling Zhang, Saijuan Liu, Shuangshuang Ma, Yan Ma, and Chi Wai Ng for their excellent technical assistance, as well as other members of the Ip laboratory for many helpful discussions. This study was supported in part by the National Basic Research Program of China (973 Program; 2013CB530900), the Hong Kong Research Grants Council Theme-based Research Scheme (T13-607/12R), the General Research Fund (grant number GRF CUHK 471911), the Area of Excellence Scheme of the University Grants Committee (AoE/M-604/16), Innovation Technology Commission (ITS/393/15FP and ITCPD/17-9), the National Natural Science Foundation of China (31671047 and 31400923), the National Key R&D Program of China (SQ2018YFE020417, 2017YFE0190000), the Guangdong Provincial Key S&T Program (2018B030336001), and the Shenzhen Knowledge Innovation Program (JCYJ20151030140325152, JCYJ20170413173717055, JCYJ20151030154629774, JCYJ20170413165053031, and JCYJ20160428145818099). X.Z. was a recipient of the Hing Kee Java Edible Bird's Nest (HKJEBN) Company Limited Scholarship for Health and Quality Living. Please refer to the Supplementary Notes 1 and 2 for corresponding acknowledgments for the ADNI dataset, Alzheimer's Disease Genetics Consortium (ADGC) Genome Wide Association Study–NIA Alzheimer's Disease Centers Cohort (ADC dataset, the National Institute on Aging–Late Onset Alzheimer's Disease Family Study (LOAD dataset), funding support for the "Genetic Consortium for Late Onset Alzheimer's Disease", the Genotype-Tissue Expression (GTEx) Project and the CommonMind dataset. Part of the data used in the preparation of this article were obtained from the Alzheimer's Disease Neuroimaging Initiative (ADNI) database (adni.loni.usc.edu). As such, the investigators within ADNI contributed to the design and implementation of ADNI and/or provided data but did not participate in the analysis or writing of this report. A complete listing of ADNI investigators can be found in the Supplementary Note and at the following URL: [http://adni.loni.usc.edu/wp-content/uploads/how_to_apply/ADNI_Acknowledgement_List.pdf].

## Author contributions

X.Z., K.Y.M., A.K.F., Y.L. and N.Y.I. conceived of the project; Y.C., T.C.K., V.C.M., Q.G. and F.C.I. organized patient recruitment and sample collection; X.Z., Y.C., Y-W.C. and N.M. performed the experiments; X.Z., K.Y.M. and Y.L. set up the data-processing pipelines; X.Z., Y.C., K.Y.M., J.H., Y.L, A.K.F and N.Y.I. analyzed the data; X.Z., A.K.F., Y.L. and N.Y.I. wrote the manuscript; P.G-R., P.F.S. and the Alzheimer's Disease Neuroimaging Initiative contributed part of the study data.

## Additional information

**Competing interests:** The authors declare no competing interests.

## Alzheimer's Disease Neuroimaging Initiative

Michael W. Weiner[12], Paul Aisen[13], Ronald Petersen[14], Clifford R. Jack[14], William Jagust[15], John Q. Trojanowski[16], Arthur W. Toga[17], Laurel Beckett[18], Robert C. Green[19], Andrew J. Saykin[20], John Morris[21], Leslie M. Shaw[16], Zaven Khachaturian[18,22], Greg Sorensen[23], Lew Kuller[24], Marcus Raichle[21], Steven Paul[25], Peter Davies[26], Howard Fillit[27], Franz Hefti[28], David Holtzman[21], Marek M. Mesulam[29], William Potter[30], Peter Snyder[31], Adam Schwartz[32], Tom Montine[33], Ronald G. Thomas[33], Michael Donohue[33], Sarah Walter[33], Devon Gessert[33], Tamie Sather[33], Gus Jiminez[33], Danielle Harvey[18], Matthew Bernstein[14], Paul Thompson[34], Norbert Schuff[12,18], Bret Borowski[14], Jeff Gunter[14], Matt Senjem[14], Prashanthi Vemuri[14], David Jones[14], Kejal Kantarci[14], Chad Ward[14], Robert A. Koeppe[35], Norm Foster[36], Eric M. Reiman[37], Kewei Chen[37], Chet Mathis[27], Susan Landau[15], Nigel J. Cairns[21], Erin Householder[21], Lisa Taylor-Reinwald[21], Virginia Lee[16], Magdalena Korecka[16], Michal Figurski[16], Karen Crawford[17], Scott Neu[17], Tatiana M. Foroud[20], Steven G. Potkin[38], Li Shen[20], Kelley Faber[20], Sungeun Kim[20], Kwangsik Nho[20], Leon Thal[13], Neil Buckholtz[39], Marylyn Albert[40], Richard Frank[41], John Hsiao[39], Jeffrey Kaye[42], Joseph Quinn[42], Betty Lind[42], Raina Carter[42], Sara Dolen[42], Lon S. Schneider[17], Sonia Pawluczyk[17], Mauricio Beccera[17], Liberty Teodoro[17], Bryan M. Spann[17], James Brewer[13], Helen Vanderswag[13], Adam Fleisher[13,37], Judith L. Heidebrink[35], Joanne L. Lord[35], Sara S. Mason[14], Colleen S. Albers[14], David Knopman[14], Kris Johnson[14], Rachelle S. Doody[43], Javier Villanueva-Meyer[43], Munir Chowdhury[43], Susan Rountree[43], Mimi Dang[43], Yaakov Stern[43], Lawrence S. Honig[43], Karen L. Bell[43], Beau Ances[21], Maria Carroll[21], Sue Leon[21], Mark A. Mintun[21], Stacy Schneider[21], Angela Oliver[21], Daniel Marson[44], Randall Griffith[44], David Clark[44], David Geldmacher[44], John Brockington[44], Erik Roberson[44], Hillel Grossman[45], Effie Mitsis[45], Leyla de Toledo-Morrell[46], Raj C. Shah[46], Ranjan Duara[47], Daniel Varon[47], Maria T. Greig[47], Peggy Roberts[47], Chiadi Onyike[40], Daniel D'Agostino[40], Stephanie Kielb[40], James E. Galvin[48], Brittany Cerbone[48], Christina A. Michel[48], Henry Rusinek[48], Mony J. de Leon[48], Lidia Glodzik[48], Susan De Santi[48], P Murali Doraiswamy[49], Jeffrey R. Petrella[49], Terence Z. Wong[49], Steven E. Arnold[16], Jason H. Karlawish[16], David Wolk[16], Charles D. Smith[50], Greg Jicha[50], Peter Hardy[50], Partha Sinha[50], Elizabeth Oates[50], Gary Conrad[50], Oscar L. Lopez[24], MaryAnn Oakley[24], Donna M. Simpson[40], Anton P. Porsteinsson[51], Bonnie S. Goldstein[52], Kim Martin[52], Kelly M. Makino[52], M Saleem Ismail[52], Connie Brand[52], Ruth A. Mulnard[38], Gaby Thai[38], Catherine McAdams-Ortiz[38], Kyle Womack[52], Dana Mathews[52], Mary Quiceno[52], Ramon Diaz-Arrastia[52], Richard King[52], Myron Weiner[52], Kristen Martin-Cook[52], Michael DeVous[52], Allan I Levey[53], James J. Lah[53], Janet S. Cellar[53], Jeffrey M. Burns[54], Heather S. Anderson[54], Russell H. Swerdlow[54], Liana Apostolova[34], Kathleen Tingus[34], Ellen Woo[34], Daniel H.S. Silverman[34], Po H. Lu[34], George Bartzokis[34], Neill R. Graff-Radford[55], Francine Parfitt[55], Tracy Kendall[55], Heather Johnson[55], Martin R. Farlow[20], Ann Marie Hake[20], Brandy R. Matthews[20], Scott Herring[20], Cynthia Hunt[20], Christopher H. van Dyck[56], Richard E. Carson[56], Martha G. MacAvoy[56], Howard Chertkow[57], Howard Bergman[57], Chris Hosein[57], Ging-Yuek Robin Hsiung[58], Howard Feldman[58], Benita Mudge[58], Michele Assaly[58], Charles Bernick[59], Donna Munic[59], Andrew Kertesz[60], John Rogers[60], Dick Trost[60], Diana Kerwin[29], Kristine Lipowski[29], Chuang-Kuo Wu[29], Nancy Johnson[29], Carl Sadowsky[61], Walter Martinez[61], Teresa Villena[61], Raymond Scott Turner[62], Kathleen Johnson[62], Brigid Reynolds[62], Reisa A. Sperling[19], Keith A. Johnson[19], Gad Marshall[19], Meghan Frey[19], Barton Lane[19], Allyson Rosen[19], Jared Tinklenberg[19], Marwan N. Sabbagh[63], Christine M. Belden[63], Sandra A. Jacobson[63], Sherye A. Sirrel[63], Neil Kowall[63], Ronald Killiany[64], Andrew E. Budson[64], Alexander Norbash[64], Patricia Lynn Johnson[64], Joanne Allard[65], Alan Lerner[66], Paula Ogrocki[66], Leon Hudson[66], Evan Fletcher[18], Owen Carmichael[18], John Olichney[18], Charles DeCarli[18],

Smita Kittur[67], Michael Borrie[68], T-Y. Lee[68], Rob Bartha[68], Sterling Johnson[69], Sanjay Asthana[69], Cynthia M. Carlsson[69], Adrian Preda[34], Dana Nguyen[34], Pierre Tariot[36], Stephanie Reeder[36], Vernice Bates[70], Horacio Capote[70], Michelle Rainka[70], Douglas W. Scharre[71], Maria Kataki[71], Anahita Adeli[71], Earl A. Zimmerman[72], Dzintra Celmins[72], Alice D. Brown[72], Godfrey D. Pearlson[73], Karen Blank[73], Karen Anderson[73], Robert B. Santulli[74], Tamar J. Kitzmiller[74], Eben S. Schwartz[74], Kaycee M. Sink[75], Jeff D. Williamson[75], Pradeep Garg[75], Franklin Watkins[75], Brian R. Ott[76], Henry Querfurth[76], Geoffrey Tremont[76], Stephen Salloway[77], Paul Malloy[77], Stephen Correia[77], Howard J. Rosen[12], Bruce L. Miller[12], Jacobo Mintzer[78], Kenneth Spicer[78], David Bachman[78], Stephen Pasternak[60], Irina Rachinsky[60], Dick Drost[60], Nunzio Pomara[79], Raymundo Hernando[79], Antero Sarrael[79], Susan K. Schultz[80], Laura L. Boles Ponto[80], Hyungsub Shim[80], Karen Elizabeth Smith[80], Norman Relkin[25], Gloria Chaing[25], Lisa Raudin[22,25], Amanda Smith[81], Kristin Fargher[81], Balebail Ashok Raj[81], Thomas Neylan[12], Jordan Grafman[29], Melissa Davis[13], Rosemary Morrison[13], Jacqueline Hayes[12], Shannon Finley[12], Karl Friedl[82], Debra Fleischman[46], Konstantinos Arfanakis[46], Olga James[49], Dino Massoglia[78], J Jay Fruehling[69], Sandra Harding[69], Elaine R. Peskind[33], Eric C. Petrie[71], Gail Li[71], Jerome A. Yesavage[83], Joy L. Taylor[83] & Ansgar J. Furst[83]

[12]UC San Francisco, San Francisco, CA 94143, USA. [13]UC San Diego, San Diego, CA 92093, USA. [14]Mayo Clinic, Rochester, NY 14603, USA. [15]UC Berkeley, Berkeley, CA 94720, USA. [16]UPenn, Philadelphia, PA 9104, USA. [17]USC, Los Angeles, CA 90089, USA. [18]UC Davis, Davis, CA 95616, USA. [19]Brigham and Women's Hospital/Harvard Medical School, Boston, MA 02115, USA. [20]Indiana University, Bloomington, IN 47405, USA. [21]Washington University in St Louis, St Louis, MI 63130, USA. [22]Prevent Alzheimer's Disease 2020, Rockville, MD 20850, USA. [23]Siemens, 80333 Munich, Germany. [24]University of Pittsburgh, Pittsburgh, PA 15260, USA. [25]Weill Cornell Medical College, Cornell University, New York City, NY 10065, USA. [26]Albert Einstein College of Medicine of Yeshiva University, Bronx, NY 10461, USA. [27]AD Drug Discovery Foundation, New York City, NY 10019, USA. [28]Acumen Pharmaceuticals, Livermore, CA 94551, USA. [29]Northwestern University, Evanston and Chicago, Evanston, IL 60208, USA. [30]National Institute of Mental Health, Rockville, MD 20852, USA. [31]Brown University, Providence, RI 02912, USA. [32]Eli Lilly, Indianapolis, IN 46225, USA. [33]University of Washington, Seattle, WA 98195, USA. [34]UCLA, Los Angeles, CA 90095, USA. [35]University of Michigan, Ann Arbor, MI 48109, USA. [36]University of Utah, Salt Lake City, UT 84112, USA. [37]Banner Alzheimer's Institute, Phoenix, AZ 85006, USA. [38]UC Irvine, Irvine, CA 92697, USA. [39]National Institute on Aging, Bethesda, MD 20892, USA. [40]Johns Hopkins University, Baltimore, MD 21218, USA. [41]Richard Frank Consulting, Washington 20001, USA. [42]Oregon Health and Science University, Portland, OR 97239, USA. [43]Baylor College of Medicine, Houston, TX 77030, USA. [44]University of Alabama, Birmingham, AL 35233, USA. [45]Mount Sinai School of Medicine, New York City, NY 10029, USA. [46]Rush University Medical Center, Chicago, IL 60612, USA. [47]Wien Center, Miami, FL 33140, USA. [48]New York University, New York City, NY 10003, USA. [49]Duke University Medical Center, Durham, NC 27710, USA. [50]University of Kentucky, Lexington, KY 0506, USA. [51]University of Rochester Medical Center, Rochester, NY 14642, USA. [52]University of Texas Southwestern Medical School, Dallas, TX 75390, USA. [53]Emory University, Atlanta, GA 30322, USA. [54]Medical Center, University of Kansas, Kansas City, KS 66103, USA. [55]Mayo Clinic, Jacksonville, FL 32224, USA. [56]Yale University School of Medicine, New Haven, CT 06510, USA. [57]McGill University/Montreal-Jewish General Hospital, Montreal, QC H3T 1E2, Canada. [58]University of British Columbia Clinic for AD and Related Disorders, Vancouver, BC V6T 1Z3, Canada. [59]Cleveland Clinic Lou Ruvo Center for Brain Health, Las Vegas, NV 89106, USA. [60]St Joseph's Health Care, London, ON N6A 4V2, Canada. [61]Premiere Research Institute, Palm Beach Neurology, Miami, FL 33407, USA. [62]Georgetown University Medical Center, Washington, DC 20007, USA. [63]Banner Sun Health Research Institute, Sun City, AZ 85351, USA. [64]Boston University, Boston, MA 02215, USA. [65]Howard University, Washington, DC 20059, USA. [66]Case Western Reserve University, Cleveland, OH 20002, USA. [67]Neurological Care of CNY, Liverpool, NY 13088, USA. [68]Parkwood Hospital, London, ON N6C 0A7, Canada. [69]University of Wisconsin, Madison, WI 53706, USA. [70]Dent Neurologic Institute, Amherst, NY 14226, USA. [71]Ohio State University, Columbus, OH 43210, USA. [72]Albany Medical College, Albany, NY 12208, USA. [73]Olin Neuropsychiatry Research Center, Hartford Hospital, Hartford, CT 06114, USA. [74]Dartmouth-Hitchcock Medical Center, Lebanon, NH 03766, USA. [75]Wake Forest University Health Sciences, Winston-Salem, NC 27157, USA. [76]Rhode Island Hospital, Providence, RI 02903, USA. [77]Butler Hospital, Providence, RI 02906, USA. [78]Medical University South Carolina, Charleston, SC 29425, USA. [79]Nathan Kline Institute, Orangeburg, NY 10962, USA. [80]University of Iowa College of Medicine, Iowa City, IA 52242, USA. [81]USF Health Byrd Alzheimer's Institute, University of South Florida, Tampa, FL 33613, USA. [82]Department of Defense, Arlington, VA 22350, USA. [83]Stanford University, Stanford, CA 94305, USA

