## [Peer Review File · Nature Communications]

Editorial Note: Parts of this Peer Review File have been redacted as indicated to maintain the confidentiality of comments made by one of the reviewers.

Reviewers' comments:

Reviewer #1 (Remarks to the Author):

In this study Zhou et al use existing and newly-recruited patient cohorts to identify new genetic variants associated with Alzheimer's Disease (AD), specifically by WGS fine-mapping of the APOE locus, leading to the identification of risk haplotypes. The new risk haplotypes in the PVRL2 and APOC1 regions confer risk independently of the APOE- ϵ 4 genotype and are associated with specific brain and systemic phenotypes: decreased cognitive performance (PVRL2 haplotype alpha), decreased brain volume (PVRL2 haplotype alpha), reduced CSF ICAM-1 levels (PVRL2 haplotype alpha), reduced total A β 1–42 in plasma (PVRL2 haplotype alpha), reduced free plasma A β 1–40 and MCP3 levels (APOC1 haplotype gamma), and increased brain APOE and APOC1 expression (PVRL2 haplotype alpha and APOC1 haplotype gamma). In addition, the authors show using available brain chromatin interaction datasets (HiC), that the PVRL2 risk haplotype region interacts with the PVRL2 promoter in adult, but not fetal brain and the APOC1 risk haplotype region interacts with the PVRL2-TOMM40 region in both adult and fetal brain.

The manuscript is well written and the figures are generally clear and attractive. The new findings may suggest additional genes and pathways in the pathophysiology of AD, as well as clarify disease risk/stratification and would be of interest to the field of neurodegeneration. My main concern is that there is no functional validation of any of the variants and open questions about their regulatory role, which somewhat limits the impact of the study. In addition, the chromatin interaction analysis seems to have been limited to the nearby locus, missing an opportunity to understand the potential role of risk haplotype regions more broadly. Please see below for more specific comments and suggestions.

1. The study would benefit from functional validation, at least proof of principle, of some of the risk variants and/or haplotypes, in particular their potential effect in the regulation of gene expression. PVRL2 haplotype alpha is associated with increased APOE expression in the brain, but shows no chromatin interaction with APOE. What is the proposed mechanism? Similarly, how does APOC1 haplotype gamma alter APOE expression? PVRL2 risk haplotypes are associated with altered PVRL2 levels in blood. At least one of these contains a variant in the 3'UTR of PVRL2, is this variant within a known microRNA binding site? How do PVRL2 risk haplotypes regulate PVRL2 gene expression?

2. Chromatin interaction analysis was restricted to the regions closest to the risk haplotypes, but chromatin interactions occur over long distances and extending such analysis may suggest additional regulatory roles for these regions.

3. Associations between the risk haplotypes and specific phenotypes are included in Figure 5, with the rest of the Information included as supplementary tables. Consider a heatmap display which would summarize the information in a more accessible manner.

Reviewer #2 (Remarks to the Author):

Comments.

=====

Zhou and colleagues examined whether genes/variants flanking the APOE gene confer independent effects on Alzheimer disease- related phenotypes in addition to the APOE E4 allele (APOE4), as multiple studies have supported the notion that there exist additional variants that explain the ‘unexplained’ phenotypic variance. The investigators carefully and thoroughly examined multiple genomic and epigenomic datasets of Chinese as well as other ancestries to test several biological properties. Below few concerns are listed.

Concerns.

- To understand the role of genes/variants flanking APOE4, independent of APOE4, it would have been better to start off with the discussion of the effects of flanking genes in APOE3 carriers, and then move onto include all subjects to discuss the interactions with flanking loci and APOE4. Although the sample size is reduced somewhat, the conclusion would have been firmer. The authors do show this sub-analysis toward the latter part of the manuscript.
- It is unclear as to how the 2 Chinese cohorts (1172 from Mainland China and 208 from Hong Kong) were selected without searching the original manuscript. How were the principal component analysis performed from the whole genome sequencing (WGS) data? It is assumed that the investigators were working with 1380 WGS data. Did the investigators include location as a covariate in their regression analysis?

- The statement on page 5 (line 87-89) – “a combination of risk alleles from multiple variants (haplotype)” -- is not convincing. A person can have the same set of genotypes in multiple contiguous genes, but different haplotypes. Such a person will have the same dosage of alleles. Unless the investigators are arguing that certain haplotypes lead to structural alterations due to a specific haplotype configuration, this argument does not hold. While it is agreed that multiple variants can together contribute to AD or related phenotypes, but why should they be on the same strand (i.e., haplotype)?

- In the association analysis, it seems to be problematic to include both APOE4 and APOE2 dosages in the same model as APOE4 and APOE2 dosages are correlated. That is, a person with a 2-dose of APOE4 will have 0-dose of APOE2 and vice versa. Do you mean that you simply used APOE genotype

- Some of the statements are speculative and can be omitted. For example, “a stronger interaction between the APOE risk haplotype region and PVRL2-TOMM40 region in the adults brain than the fetal brain, suggesting that the alter interaction between the PVRL2-TOMM40 and APOE regions in the adult brain may contribute to the functional effects of these risk variants in gene regulation.” Statements such as this need to be followed up with an experiment showing differential interactions between controls vs. MCI cases (that are in the process of dementing).

Minor

- It would be helpful to include Table 1 that describes the study populations and allele frequencies for candidate SNPs.

- Hi-C analysis in the abstract should be spelled out and explained.

Reviewers' comments:

Reviewer #1 (Remarks to the Author):

In this study Zhou et al use existing and newly-recruited patient cohorts to identify new genetic variants associated with Alzheimer's Disease (AD), specifically by WGS fine-mapping of the APOE locus, leading to the identification of risk haplotypes. The new risk haplotypes in the PVRL2 and APOC1 regions confer risk independently of the APOE- ϵ 4 genotype and are associated with specific brain and systemic phenotypes: decreased cognitive performance (PVRL2 haplotype alpha), decreased brain volume (PVRL2 haplotype alpha), reduced CSF ICAM-1 levels (PVRL2 haplotype alpha), reduced total A β 1–42 in plasma (PVRL2 haplotype alpha), reduced free plasma A β 1–40 and MCP3 levels (APOC1 haplotype gamma), and increased brain APOE and APOC1 expression (PVRL2 haplotype alpha and APOC1 haplotype gamma). In addition, the authors show using available brain chromatin interaction datasets (HiC), that the PVRL2 risk haplotype region interacts with the PVRL2 promoter in adult, but not fetal brain and the APOC1 risk haplotype region interacts with the PVRL2-TOMM40 region in both adult and fetal brain.

The manuscript is well written and the figures are generally clear and attractive. The new findings may suggest additional genes and pathways in the pathophysiology of AD, as well as clarify disease risk/stratification and would be of interest to the field of neurodegeneration. My main concern is that there is no functional validation of any of the variants and open questions about their regulatory role, which somewhat limits the impact of the study. In addition, the chromatin interaction analysis seems to have been limited to the nearby locus, missing an opportunity to understand the potential role of risk haplotype regions more broadly. Please see below for more specific comments and suggestions.

We thank the reviewer for the comment. Accordingly, we conducted additional analyses that included a database query and wet-lab experiments to demonstrate the possible biological functions of candidate risk haplotypes. We showed that specific genomic regions harboring the variants can bind to microRNAs and transcription factors. Together with the gel shift results demonstrating the binding of the candidate variants to nuclear proteins, we suggest that these genetic variants are involved in the regulation of specific gene expression. We also conducted additional analyses to examine the chromatin interaction events on broader genomic regions. The details of the newly added results are described as follows.

1. The study would benefit from functional validation, at least proof of principle, of some of the risk variants and/or haplotypes, in particular their potential effect in the regulation of gene expression. PVRL2 haplotype alpha is associated with increased APOE expression in the brain, but shows no chromatin interaction with APOE. What is the proposed mechanism? Similarly, how does APOC1 haplotype gamma alter APOE expression? PVRL2 risk haplotypes are associated with altered PVRL2 levels in blood. At least one of these contains a variant in the 3'UTR of PVRL2, is this variant within a known microRNA binding site? How do PVRL2 risk haplotypes regulate PVRL2 gene expression?

We conducted an additional database analysis on the genomic regions that harbor corresponding haplotypes. For example, the chromatin immunoprecipitation (ChIP) signals of histone H3 mono-methylation at lysine 4 (H3K4me1) and histone H3 acetylation at lysine 27 (H3K27ac) from human hippocampal tissues as well as H3K27ac ChIP data from various

human cell lines (ENCODE project) might reveal genomic regions with potential enhancer activity. The analysis showed that various risk variants of the identified haplotypes (e.g., rs6859 and rs11568822) are located in genomic regions that are associated with transcription factor binding and histone modification (**Supplementary Figure 13**). This suggests a possible mechanism through which the risk haplotype(s) exert their risk effect by modulating the transcriptional activities of specific genes. Furthermore, we conducted an electrophoretic mobility shift assay and showed that specific genomic regions including rs6859 and rs11568822 can bind to nuclear proteins (**Supplementary Figure 14**). Thus, the corresponding genomic regions might exert functions through binding with nuclear factors, possibly transcription factors. For rs6859 in the untranslated regions that are associated with alteration of *PVRL2* gene expression, *in silico* analysis suggested that microRNA miR-595 might interact with rs6859 in an allele-specific manner (**Supplementary Tables 45–47**). We included these results in the main text under the section entitled “*Functional implications of non-coding variants in association with enhancer activity and microRNA binding*” (main text **lines 531–549**).

2. Chromatin interaction analysis was restricted to the regions closest to the risk haplotypes, but chromatin interactions occur over long distances and extending such analysis may suggest additional regulatory roles for these regions.

We thank the reviewer for the comment. As suggested, we conducted an additional analysis and discovered a boarder genomic region that physically interacted with the extended haplotype regions in both fetal and adult brain (**Supplementary Figures 11, 12**; main text **lines 522–523**).

3. Associations between the risk haplotypes and specific phenotypes are included in Figure 5, with the rest of the Information included as supplementary tables. Consider a heatmap display which would summarize the information in a more accessible manner.

We thank the reviewer for the insightful suggestions. We summarized the associations test results as heatmaps with colors (blue and red) indicating the trend of effect size and shading for *p*-values (in \log_{10} scale) (**Supplementary Figure 3**; main text **lines 422–423**).

Reviewer #2 (Remarks to the Author):

Comments.

=====
Zhou and colleagues examined whether genes/variants flanking the APOE gene confer independent effects on Alzheimer disease-related phenotypes in addition to the APOE E4 allele (APOE4), as multiple studies have supported the notion that there exist additional variants that explain the ‘unexplained’ phenotypic variance. The investigators carefully and thoroughly examined multiple genomic and epigenomic datasets of Chinese as well as other ancestries to test several biological properties. Below few concerns are listed.

Concerns.

- To understand the role of genes/variants flanking APOE4, independent of APOE4, it would have been better to start off with the discussion of the effects of flanking genes in APOE3 carriers, and then move onto include all subjects to discuss the interactions with flanking loci and APOE4. Although the sample size is reduced somewhat, the conclusion would have been firmer. The authors do show this sub-analysis toward the latter part of the manuscript.

We thank the reviewer for the suggestions and have made revisions accordingly (**new Figure 1**; main text **lines 279–297**).

- It is unclear as to how the 2 Chinese cohorts (1172 from Mainland China and 208 from Hong Kong) were selected without searching the original manuscript. How were the principal component analysis performed from the whole genome sequencing (WGS) data? It is assumed that the investigators were working with 1380 WGS data. Did the investigators include location as a covariate in their regression analysis?

We thank the reviewer for the comment. The WGS analysis and variant calling were conducted separately using the Gotcloud pipeline, with genotyping refinement performed by Beagle. The principal components analysis was conducted by including pruned SNPs with an MAF > 5% separately for the mainland and Hong Kong cohorts. Statistical analysis was conducted separately for the mainland and Hong Kong datasets, with the top five principal components included to control for population structure. Accordingly, we included this information in the revised manuscript (main text **lines 116–123, 169–173**).

- The statement on page 5 (line 87-89) – “a combination of risk alleles from multiple variants (haplotype)” -- is not convincing. A person can have the same set of genotypes in multiple contiguous genes, but different haplotypes. Such a person will have the same dosage of alleles. Unless the investigators are arguing that certain haplotypes lead to structural alterations due to a specific haplotype configuration, this argument does not hold. While it is agreed that multiple variants can together contribute to AD or related phenotypes, but why should they be on the same strand (i.e., haplotype)?

We thank the reviewer for the careful review. We deleted the word “haplotype” from the captioned text (main text **lines 87–89**).

- In the association analysis, it seems to be problematic to include both APOE4 and APOE2 dosages in the same model as APOE4 and APOE2 dosages are correlated. That is, a person with a 2-dose of APOE4 will have 0-dose of APOE2 and vice versa. Do you mean that you

simply used APOE genotype

We thank the reviewer for the comment. Given that *APOE* has three isoforms in the general population—E2, E3 (the most common), and E4—the E4 and E2 isoforms are not perfectly correlated. Moreover, E4 is a risk factor, whereas E2 has a protective effect against AD. Thus, the inclusion of the E4 and E2 genotypes as covariates can control for their possible confounding effects.

- Some of the statements are speculative and can be omitted. For example, “a stronger interaction between the APOE risk haplotype region and PVRL2-TOMM40 region in the adults brain than the fetal brain, suggesting that the alter interaction between the PVRL2-TOMM40 and APOE regions in the adult brain may contribute to the functional effects of these risk variants in gene regulation.” Statements such as this need to be followed up with an experiment showing differential interactions between controls vs. MCI cases (that are in the process of dementing).

We thank the reviewer for the thoughtful comment. We revised the text by deleting the speculative statements (including the aforementioned paragraph) and added the following statement to increase the urgency for a follow-up study (main text **lines 515–517, 525–528**):
“These observations suggest the complexity of chromatin states that might contribute to the regulation of transcriptional activity, prompted the urgency for the further investigation of associated chromatin structure changes in the brain during the aging or dementing stage.”

Minor

- It would be helpful to include Table 1 that describes the study populations and allele frequencies for candidate SNPs.

We thank the reviewer for the comment. Regarding Table 1, we previously included the frequencies of causal variants from the mainland Chinese normal controls. Accordingly, we updated the table by adding the corresponding frequencies from other study cohorts (i.e., the Hong Kong AD, ADNI, ADC, and LOAD cohorts) (**updated Table 1**).

- Hi-C analysis in the abstract should be spelled out and explained.

We revised “Hi-C analysis” to “high-throughput chromosome conformation capture analysis” in the abstract (main text **lines 58–59**).

Reviewers' comments:

Reviewer #1 (Remarks to the Author):

In this revision Zhou et al. provide new experiments and database queries to support the interpretation of how several AD risk variants may impact gene regulation. They also extend Hi-C analysis to a broader region to show extensive contacts within 1Mb. Although a step in the right direction, these experiments fall short of providing sufficient evidence to support the conclusions or reveal regulatory mechanisms. The authors use EMSA to show binding of nuclear proteins to two candidate variant regions. As performed, this assay is not compelling. The extract is from HEK cells which is not a physiologically relevant cell type and provides no specificity as to which nuclear factors may be binding to the probes. Through their database query, the authors identified several TFs which may be predicted to bind in the loci containing the variants. The better experiment would be to show if these TFs bind to the probes and if binding is altered by the presence of the major or minor allele. The authors further observe that a risk locus in the 3'UTR of PVRL2 has predicted binding to mir-595 and that this predicted interaction is different between the major and minor alleles. This is another prediction, rather than functional evidence. The authors should follow this in silico analysis with luciferase reporter experiments showing modulation of a reporter containing this 3' UTR in the presence or absence of mir-595, including the major and minor alleles. These studies would provide proof of principle validation of the mechanism by which these variants may relate to phenotypic observations.

Reviewer #2 (Remarks to the Author):

Zhou and colleagues responded to all of my queries satisfactorily. This study is well-done and provides a possible explanation for variable genotype-phenotype relations in the APOE region observed in the literature.

Point-by-point responses:

Reviewer #1 (Remarks to the Author):

In this revision Zhou et al. provide new experiments and database queries to support the interpretation of how several AD risk variants may impact gene regulation. They also extend Hi-C analysis to a broader region to show extensive contacts within 1Mb. Although a step in the right direction, these experiments fall short of providing sufficient evidence to support the conclusions or reveal regulatory mechanisms. The authors use EMSA to show binding of nuclear proteins to two candidate variant regions. As performed, this assay is not compelling. The extract is from HEK cells which is not a physiologically relevant cell type and provides no specificity as to which nuclear factors may be binding to the probes. Through their database query, the authors identified several TFs which may be predicted to bind in the loci containing the variants. The better experiment would be to show if these TFs bind to the probes and if binding is altered by the presence of the major or minor allele.

The authors further observe that a risk locus in the 3'UTR of PVRL2 has predicted binding to mir-595 and that this predicted interaction is different between the major and minor alleles. This is another prediction, rather than functional evidence. The authors should follow this in silico analysis with luciferase reporter experiments showing modulation of a reporter containing this 3' UTR in the presence or absence of mir-595, including the major and minor alleles. These studies would provide proof of principle validation of the mechanism by which these variants may relate to phenotypic observations.

We thank the reviewer for suggesting two experiments to further examine the functional effects of the observed allele-specific risk variants, including the modulation of the binding affinity of transcription factors and the *in vitro* regulation of gene expression by microRNAs. However, we wish to emphasize that the key point of our study is the identification of non-coding AD risk haplotypes, which represent joint risk effects from multiple variants, in the region near *APOE*. Our results indeed demonstrate the presence of AD risk signals in *APOE* and its nearby region that are independent of *APOE*- ϵ 4. This is a very important finding, because it resolves the question of whether there are non-coding AD risk signals in the *APOE* region.

Accordingly, we received a very positive comment from the same reviewer in the previous round of review: “The manuscript is well written and the figures are

generally clear and attractive. The new findings may suggest additional genes and pathways in the pathophysiology of AD, as well as clarify disease risk/stratification and would be of interest to the field of neurodegeneration.”

To address the reviewer’s comment (in the first round of review) about the lack of functional validation for the identified risk haplotypes, we conducted additional experiments and an *in silico* analysis as proof-of-concept studies, and included the results in the revised manuscript previously submitted. The results demonstrate that risk variant-associated genomic regions bind with nuclear proteins and also suggest a possible allele-specific interaction between miR-595 and the *PVRL2* UTR region (which harbors the risk variant rs6859). Thus, our results show that these are possible mechanisms by which the risk haplotypes contribute to AD.

The two experiments proposed by the reviewer in the second round of review can only examine the risk effects of individual variants rather than the whole haplotype structures under non-physiological conditions; thus, they are similar to our existing proof-of-concept work. It is also unclear what specific cell types should be used for the functional study. APOE is expressed essentially in astrocytes in the resting brains, whereas its expression increases massively in microglia of AD brains¹⁻³. To have a better understanding of the biological roles of the haplotypes, we would need to take into consideration of the variability of different cell types under both resting and pathological conditions. Moreover, as our major findings concern AD risk haplotypes rather than individual risk variants, a more physiologically relevant system (e.g., an induced pluripotent stem cell-derived system that harbors the risk haplotypes) would be required to examine the regulatory mechanism that underlies the identified risk haplotypes. However, this study is well beyond the scope of our study.

1. Huang, Y., Weisgraber, K.H., Mucke, L. & Mahley, R.W. Apolipoprotein E. *Journal of Molecular Neuroscience* 23, 189-204 (2004).
2. Krasemann, S. et al. The TREM2-APOE Pathway Drives the Transcriptional Phenotype of Dysfunctional Microglia in Neurodegenerative Diseases. *Immunity* 47, 566-581 e9 (2017).
3. Aoki, K. et al. Increased expression of neuronal apolipoprotein E in human brain with cerebral infarction. *Stroke* 34, 875-80 (2003).

[Redacted]